# Critique-Guided Distillation for Robust Reasoning via Refinement

**Berkcan Kapusuzoglu** [1]  **Supriyo Chakraborty** [1]  **Zain Sarwar** [2]  **Chia-Hsuan Lee** [1]  **Sambit Sahu** [1]

## Abstract

Supervised fine-tuning with expert demonstrations often produces models that imitate outputs without internalizing the reasoning processes needed for robust generalization. While critique-based approaches show promise, training models to generate critiques directly, such as Critique Fine-Tuning (CFT), can lead to output-format drift and degradation of general capabilities. We propose CRITIQUE-GUIDED DISTILLATION (CGD), a training framework that decouples critique consumption from critique generation. During fine-tuning, the student is trained to refine flawed responses conditioned on teacher critiques. CGD treats critiques as a *training-time-only* supervision signal, encouraging internalization of error-aware reasoning: critiques guide learning but are absent at inference. Controlled ablations confirm that these reasoning gains are directly driven by the specificity and relevance of the teacher's feedback. Across five model families, CGD consistently outperforms CFT and standard distillation on mathematical reasoning benchmarks, yielding 7% average improvements and gains of up to +15.0% on AMC23 and +12.2% on MATH-500. On challenging competition problems such as AIME24 and AIME25, CGD achieves substantially higher Pass@1 and stronger performance at low Pass@k, indicating improved reasoning quality per sample. Importantly, CGD preserves general instruction-following capabilities where CFT degrades significantly ($-21.3\%$ on IFEval). These results position CGD as a practical and compute-efficient intermediate training paradigm for reasoning-centric tasks without introducing architectural inference-time overhead.

[1]Capital One, AI Foundations, McLean, VA 22102, USA [2]University of Chicago, Department of Computer Science, Chicago, USA. Correspondence to: Berkcan Kapusuzoglu <berkcan.kapusuzoglu@capitalone.com>.

*Proceedings of the $43^{rd}$ International Conference on Machine Learning*, Seoul, South Korea. PMLR 306, 2026. Copyright 2026 by the author(s).

## 1 Introduction

Supervised fine-tuning (SFT) is the dominant approach for adapting large language models to downstream tasks (Wei et al., 2022; Sanh et al., 2022). Despite its success, vanilla SFT exhibits well-documented limitations: increased hallucination tendency (Gekhman et al., 2024), poor out-of-distribution generalization (Chu et al., 2025), and difficulty scaling to harder problem instances (Sun et al., 2024; 2025). These shortcomings stem from a fundamental issue: SFT teaches models *what* to output without explaining *why*.

An alternative paradigm leverages critique and revision: models generate initial answers, critique them, and refine outputs based on that critique (Kim et al., 2023; Madaan et al., 2023; Shinn et al., 2023; Saunders et al., 2022). While effective, multi-pass prompting methods incur substantial inference costs. Critique Fine-Tuning (CFT) (Wang et al., 2025b) moves critique generation into training, achieving improvements over SFT on mathematical reasoning. However, training models to *produce* critiques introduces new challenges, including output-format drift and catastrophic forgetting of general capabilities, with substantial degradation on instruction-following benchmarks.

In this work, we introduce **C**ritique-**G**uided **D**istillation (CGD), illustrated in Figure 1. Rather than training models to generate critiques, CGD trains models to *use* them. Given a prompt, a student's initial (potentially incorrect) answer, and a teacher-generated critique explaining the errors, the student is trained to produce a refined solution. Crucially, critiques are used only during training. At inference time, the model generates refined answers directly from prompts in a single pass, with no additional architectural overhead.

This design reflects a key conceptual contribution: *decoupling critique consumption from critique generation*. Prior methods either require the student to generate critiques during training (CFT, causing format drift) or to consume them at inference time (Self-Refine, incurring latency). CGD treats critiques as a purely training-time semantic scaffold—the student internalizes error-correction reasoning without ever producing or receiving critiques during deployment.

Across multiple model families and benchmarks, CGD consistently outperforms standard distillation and critique-generation baselines. On challenging mathematical reason-

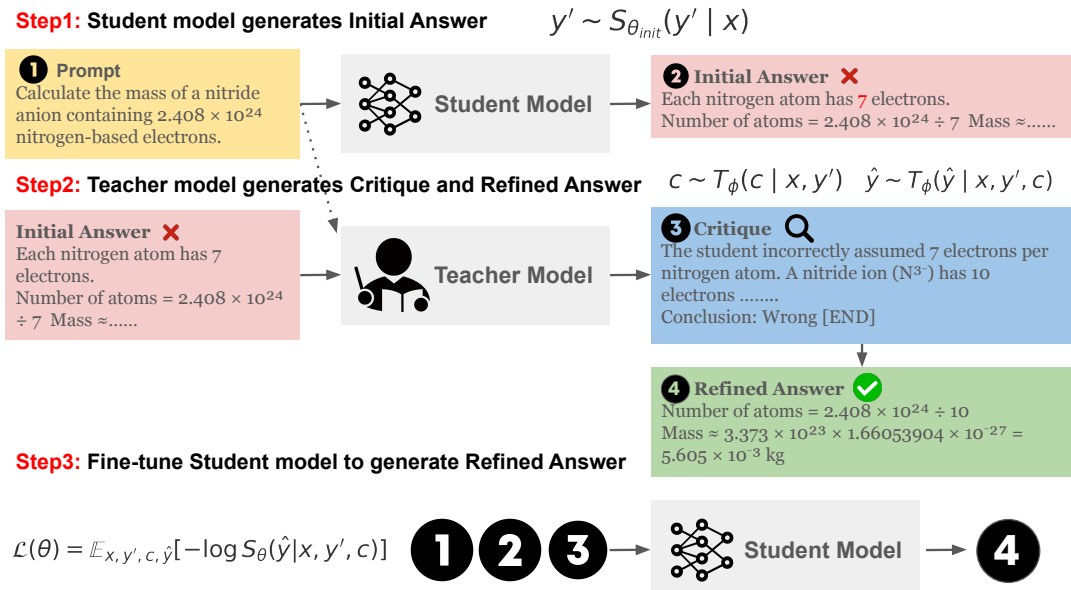

**Step1: Student model generates Initial Answer** $y' \sim S_{\theta_{init}}(y' \mid x)$

**Step2: Teacher model generates Critique and Refined Answer** $c \sim T_\phi(c \mid x, y') \quad \hat{y} \sim T_\phi(\hat{y} \mid x, y', c)$

**Step3: Fine-tune Student model to generate Refined Answer**

$\mathcal{L}(\theta) = \mathbb{E}_{x, y', c, \hat{y}}[-\log S_\theta(\hat{y}|x, y', c)]$

*Figure 1.* **Overview of Critique-Guided Distillation (CGD)**. (1) A prompt is sampled. (2) The student generates an initial response. (3) The teacher produces a critique identifying errors and (4) a refined answer. The student is fine-tuned to map (prompt, student answer, critique) → refined answer. At inference, only the prompt is provided, and the model directly outputs refined answers in one pass.

ing tasks, CGD achieves large absolute gains over CFT, including improvements of up to +15.0% on AMC23 and +12.2% on MATH-500. Figure 2 shows representative results for LLaMA3.1-8B Instruct, where CGD improves performance across all evaluated tasks relative to Distilled SFT and CFT. Importantly, unlike CFT, CGD preserves general capabilities, maintaining instruction-following performance on IFEval and transferring to out-of-distribution tasks such as code generation, despite being trained on data containing no code.

Beyond LLaMA, CGD generalizes across model families. Cross-family experiments on Qwen2.5-Math-7B (Table 3) demonstrate gains of +22.6% over the base model with only 8 GPU-hours of training, highlighting CGD's effectiveness as a practical and efficient paradigm. On challenging benchmarks such as AIME, CGD yields its largest gains at low sampling budgets, while Pass@k performance continues to improve with additional samples, indicating improved per-sample reasoning quality rather than reliance on search.

Our contributions are:

- We introduce **C**ritique-**G**uided **D**istillation (CGD), a critique-conditioned fine-tuning framework that trains models to refine their own imperfect responses using explanatory feedback, requiring only single-pass inference with no additional overhead.

- We demonstrate consistent improvements over standard distillation and CFT across five model families (LLaMA, Qwen, S1.1, Mixtral, OLMo), with gains of 5–7% on

mathematical reasoning while preserving instruction-following capabilities where CFT degrades catastrophically ($-21.3\%$ on IFEval).

- We show that CGD generalizes beyond the training domain, transferring to out-of-distribution tasks including code generation despite being trained on data without code.

- We show that CGD achieves strong cross-family generalization efficiently, offering a practical training paradigm for reasoning tasks.

- We provide empirical analysis and ablations that characterize the behavior, robustness, and limitations of CGD, clarifying when and why it is effective.

## 2 Related Work

**Limitations of Supervised Fine-Tuning.** Standard SFT trains models to mimic expert demonstrations but often induces shallow imitation without internalizing reasoning (Gekhman et al., 2024). Fine-tuned models exhibit poor out-of-distribution performance (Chu et al., 2025), and gains on training distributions often come at the cost of reliability elsewhere (Li et al., 2025). Recent work shows that SFT on reasoning traces can boost mathematical problem-solving (Muennighoff et al., 2025; Ye et al., 2025), yet models still struggle to generalize to harder instances (Sun et al., 2024; 2025). These limitations motivate approaches that go beyond answer imitation.

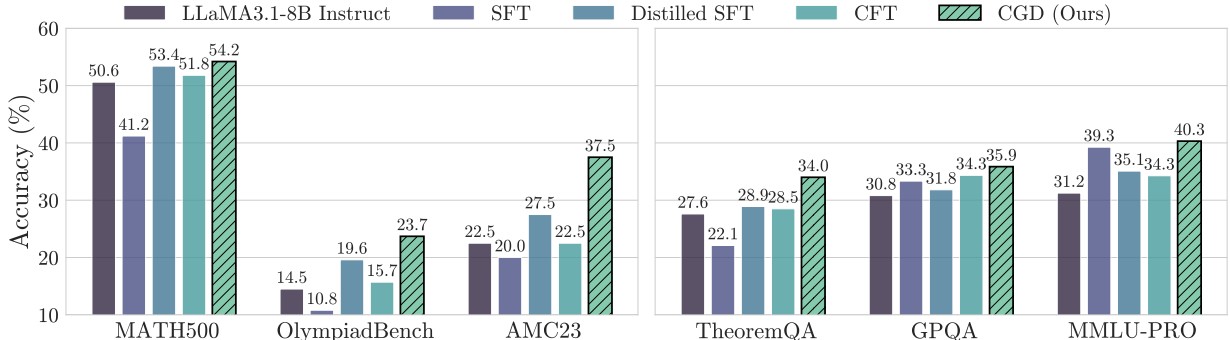

*Figure 2.* **Performance of CGD on LLaMA3.1-8B Instruct.** A LLaMA3.3-70B Instruct teacher provides critiques and refined answers for 100K WebInstruct samples. The student is trained to map from (prompt, initial answer, critique) to refined answer. CGD outperforms Distilled SFT and Critique Fine-Tuning (CFT), demonstrating the benefit of training models to act on critiques rather than generate them.

**Distillation from Reasoning Traces.** Knowledge distillation transfers capabilities from large teachers to smaller students (Hinton et al., 2015). Recent work distills not just answers but reasoning processes. ORCA (Mukherjee et al., 2023) trains students on GPT-4's step-by-step explanations; Orca-Math (Mitra et al., 2024) extends this with iterative preference learning. MoDE-CoTD (Li et al., 2024) distills chain-of-thought (CoT) via mixture-of-experts (MoE), while SCOTT (Wang et al., 2023a) uses contrastive objectives to ensure faithful rationale reproduction. These methods condition training on *generic teacher explanations* of correct solutions. CGD instead conditions on *critiques of student-specific errors*, providing feedback aligned to actual failure modes rather than idealized reasoning. Moreover, many trace-based methods implicitly encourage students to reproduce teacher-style reasoning, whereas CGD only requires producing a correct refined answer, allowing internal reasoning representations to differ.

**On-Policy Distillation.** Standard distillation is off-policy: students learn from static teacher outputs regardless of their own behavior. On-policy methods address this by training on student-generated outputs. GKD (Agarwal et al., 2024) provides teacher probability supervision on student generations. SKD (Xu et al., 2025) uses interleaved sampling to bridge teacher-student gaps. Black-box methods (Ye et al., 2026) use discriminators as adaptive rewards when teacher logits are unavailable. These approaches provide *implicit* supervision—token probabilities or scalar rewards indicating output quality without diagnosing errors. CGD provides *explicit semantic* supervision: critiques that identify specific erroneous assumptions or steps. This aligns with findings that effective feedback must be specific and actionable rather than scalar or implicit (Borges et al., 2023; Zhang et al., 2024).

**Critique-Based Learning.** Several methods leverage critiques for model improvement. Critique Fine-Tuning (CFT) (Wang et al., 2025b) trains students to *generate* critiques, improving reasoning but causing format drift and capability degradation. Prior work on self-refinement similarly emphasizes the importance of localized error explanations for targeted correction, either via uncertainty-aware signals (Zhang et al., 2024) or fine-grained constraint decomposition (Ferraz et al., 2024). Self-Refine (Madaan et al., 2023) and Reflexion (Shinn et al., 2023) implement critique-and-revise loops at inference time, incurring multiple decoding passes. Shridhar et al. (2023) and Yu et al. (2024b) similarly train models for self-correction but still require additional inference-time steps. Other approaches train separate critic models for inference-time use, such as Shepherd (Wang et al., 2023b) and CTRL (Xie et al., 2025), while RL-based methods such as RL4F (Akyürek et al., 2023) and SCoRe (Kumar et al., 2024) rely on reinforcement learning. CGD differs from all these approaches: it trains students to *consume* critiques rather than generate them, uses purely supervised learning rather than RL, and introduces no additional inference-time overhead.

**Discriminator and Latent Reasoning Methods.** Recent work explores alternative supervision signals for reasoning. GRACE (Khalifa et al., 2026) trains step-level discriminators to score reasoning chains during decoding. QCRD (Wang et al., 2025a) distills both positive and negative rationales with quality-weighted contrastive losses. CODI (Shen et al., 2025) compresses explicit reasoning into continuous latent tokens via hidden-state alignment. These methods either operate at inference time, require learned discriminators, or move reasoning into latent spaces. Unlike critiques, discriminator scores do not localize or explain errors, limiting their usefulness for targeted correction. CGD remains in discrete token space, requires no discriminator or reward model, and relies on direct teacher feedback rather than learned quality signals.

# 3 CRITIQUE-GUIDED DISTILLATION (CGD)

In this section we describe CGD and provide analysis of its training procedure.

## 3.1 Overview

The key intuition behind CGD is that a student can internalize self-correction by training on teacher-provided critiques. During training, the student sees its own errors alongside explicit feedback; at inference, it generates refined answers directly from prompts alone. This approach, summarized in Figure 1, proceeds in three stages:

1. **Student Answer Generation:** The student baseline $S_{\theta_{\text{init}}}$ produces a noisy response $y' \sim S_{\theta_{\text{init}}}(\cdot|x)$.

2. **Critique Generation:** The teacher model $T_\phi$ critiques this response, generating a textual explanation of its flaws or merits, $c \sim T_\phi(\cdot|x, y')$.

3. **Refined Answer Generation:** The teacher produces a gold-standard, refined answer $\hat{y} \sim T_\phi(\cdot|x, y', c)$, conditioned on all prior context.

Importantly, the student answer $y'$ is sampled fresh for each prompt during data generation, so critiques address the *specific errors* made by the current student model checkpoint, not generic problem solutions. This creates a curriculum naturally aligned with the student's failure modes. In contrast, standard distillation provides the same teacher solution regardless of what errors a student might make.

By conditioning answer generation on the critique, CGD avoids the format drift seen in CFT, as the model's objective remains generating answers, not critiques.

## 3.2 Training Objective

The student is fine-tuned on the augmented dataset $((x, y', c), \hat{y})$ using a standard language modeling objective. As summarized in Algorithm 1, the goal is to minimize the negative log-likelihood of the teacher's refined answer, conditioned on the full context:

$$\mathcal{L}(\theta) = \mathbb{E}_{(x, y', c, \hat{y})}[-\log S_\theta(\hat{y} \mid x, y', c)]. \quad (1)$$

**Inference Behavior.** At inference time, CGD requires only a single forward pass, making it identical in computational cost to standard SFT. Specifically, the model receives only the prompt, no student answer or critique. We do not modify the prompt format or add special tokens. The refined reasoning behavior emerges implicitly: by training on the transformation (flawed answer, critique) → refined answer, the model learns to internalize the self-correction process.

Empirically, CGD-trained models produce longer reasoning chains (4.4× on AIME, Table 5) even without explicit prompting for reflection.

---

**Algorithm 1** Critique-Guided Distillation

1: **Input:** Dataset $\mathcal{D} = \{x_i, y_i\}_{i=1}^N$, Student $S_{\theta_{init}}$, Teacher $T_\phi$
2: **Output:** Fine-tuned student $S_\theta$
3: Initialize augmented dataset $\mathcal{D}' \leftarrow \emptyset$
4: **for** each $x_i \in \mathcal{D}$ **do**
5:     Generate student answer: $y_i' \sim S_{\theta_{init}}(y|x_i)$
6:     Generate critique: $c_i \sim T_\phi(c|x_i, y_i')$
7:     Generate refined answer: $\hat{y}_i \sim T_\phi(\hat{y}|x_i, y_i', c_i)$
8:     $\mathcal{D}' \leftarrow \mathcal{D}' \cup \{(x_i, y_i', c_i, \hat{y}_i)\}$
9: **end for**
10: Train $S_\theta$ on $\mathcal{D}'$ by minimizing $\mathcal{L}(\theta)$
11: **return** $S_\theta$

---

# 4 Experiments

Our experiments are designed to show that Critique-Guided Distillation (CGD) is a highly efficient and effective method for improving the reasoning capabilities of LLMs. We show that CGD significantly outperforms strong fine-tuning baselines, including standard SFT, Distilled SFT, and CFT, across a diverse suite of challenging math and reasoning benchmarks.

Beyond accuracy, we analyze how CGD improves reasoning quality and efficiency. In particular, we demonstrate substantial gains under greedy decoding and stronger performance at low Pass@k on difficult competition benchmarks, indicating improved reasoning per sample rather than reliance on extensive sampling. We further show that CGD is robust to critique noise and hyperparameter choices, and avoids the output-format drift and capability degradation observed in prior critique-based approaches.

Together, these results support CGD as an effective and stable supervised training paradigm for learning self-correction skills that transfer to downstream reasoning tasks without introducing inference-time overhead. Additional comparisons to RL–based approaches, highlighting complementary strengths, are provided in the appendix.

## 4.1 Experimental Setup

### 4.1.1 DATASETS

We consider two datasets for training: WebInstruct (Yue et al., 2024), and MetaMathQA (Yu et al., 2024a). WebInstruct[1] is a web-crawled instruction dataset that spans a wide

---

[1]WebInstruct's collection methodology uses filtering to reduce overlap with common benchmarks. In addition, we verified no

range of topics, including Math, Physics, Chemistry, and more. MetaMathQA is a dataset based on GSM8K (Cobbe et al., 2021b) and MATH (Hendrycks et al., 2021) which synthesizes more questions and answers by rephrasing and other augmentation techniques. We randomly sample 100k examples from each dataset as training data.

We evaluate on two sets of benchmarks capturing both mathematical reasoning and broader STEM-oriented problem solving. Group 1: Math Reasoning comprises MATH500 (Hendrycks et al., 2021), Minerva-Math (Lewkowycz et al., 2022), GSM8K (Cobbe et al., 2021a), OlympiadBench (He et al., 2024), and AMC23. Group 2: General Reasoning includes TheoremQA (Chen et al., 2023), GPQA (Rein et al., 2023), and MMLU-Pro (Wang et al., 2024).

To evaluate model capabilities beyond math and science reasoning, we also report results on the following datasets to evaluate general instruction-following and question answering abilities: IFEval (Zhou et al., 2023), MUSR (Sprague et al., 2024), TruthfulQA (Lin et al., 2022), and BIG-Bench Hard (BBH) (Suzgun et al., 2022).

### 4.1.2 BASELINE AND TRAINING SETTINGS

We evaluate CGD across two student–teacher pairs to test both within-family and cross-family robustness:

- **LLaMA family:** LLaMA3.1-8B Instruct as the *student model*, and LLaMA3.3-70B Instruct as the *teacher model*.

- **Qwen family:** S1.1-3B[2] as the *student model*, and S1.1-32B[3] as the *teacher model*.

We compare CGD to three supervised fine-tuning baselines: (i) **Standard SFT:** fine-tunes the student model to generate gold answers conditioned only on the input prompt. (ii) **Distilled SFT:** fine-tunes the student to reproduce the teacher's refined answers, where each refinement is obtained by prompting the teacher with the input prompt. (iii) **Critique Fine-Tuning (CFT):** fine-tunes the student to generate the teacher-provided critiques conditioned on the input prompt and the student's initial answer.[4]

All methods are trained for exactly 1 epoch on identical 100K-example datasets with a global batch size of 64, yielding the same number of optimization steps across all methods. No method receives additional training. On 16 NVIDIA

A100 GPUs, each run completes in approximately 30 minutes (∼8 A100 GPU-hours). Identical data splits and hyperparameters are used across all methods (see Appendix A).

## 4.2 Main Results

We report the evaluation results of training on WebInstruct in Table 1. We evaluate two student–teacher pairs: LLaMA3.1-8B Instruct with LLaMA3.3-70B Instruct as the teacher, and S1.1-3B with S1.1-32B as the teacher. In addition to our CFT experiments using LLaMA3.3-70B Instruct, we also include a variant of CFT that uses 50K examples distilled with GPT-4o, sourced from Wang et al. (2025b) (denoted as *CFT\* with GPT-4o*). Cross-family validation on Qwen2.5-Math-7B is presented in Section 4.3.

A full breakdown of additional results, including ablation studies on different model architectures, teacher models, on math-specific training data (MetaMathQA), and hyperparameter sensitivity, is provided in Appendix B

CGD consistently improves over CFT across both families. On LLaMA3.1-8B, CGD achieves +5.4 average gain on math reasoning (Group 1) and +4.3 on general reasoning (Group 2), with particularly strong improvements on OlympiadBench (+8.0) and AMC23 (+15.0). On S1.1-3B, CGD achieves even larger gains of +7.2 on math and +3.5 on general reasoning, including notable improvements on MATH500 (+12.2), Minerva-Math (+6.9), and AMC23 (+7.5). These results demonstrate that critique-guided training enhances reasoning ability more broadly than CFT and distilled SFT across diverse student–teacher settings.

We additionally evaluate on out-of-distribution benchmarks (Table 2). CGD matches or surpasses all baselines, confirming that critique-conditioned training preserves or improves general capabilities including instruction-following and factual QA. By contrast, CFT's IFEval accuracy drops from 76.9% to 55.6%. We note that CFT's degradation stems from its objective (predicting critiques), not hyperparameter misconfiguration. This suggests the issue is fundamental to critique-generation objectives, which CGD avoids by training on refined answers instead of critiques.

We evaluate generalization across multiple dimensions. **Domain transfer:** Table 2 shows CGD preserves instruction-following (IFEval) and transfers to code generation (HumanEval, +4.9%) despite training data containing no code. **Task transfer:** Table 1 shows CGD improves on GPQA (graduate-level science) and TheoremQA (theorem proving), which require different reasoning patterns than WebInstruct's math focus. **Difficulty transfer:** Table 5 shows CGD achieves 5× accuracy gains on AIME 2024, competition-level problems far harder than training distribution.

To further validate domain-agnostic generalization, we eval-

---

direct overlap between WebInstruct training prompts and our evaluation benchmarks using exact string matching and fuzzy matching. We found < 0.1% potential overlap, which we removed.

[2]https://huggingface.co/simplescaling/s1.1-3B

[3]https://huggingface.co/simplescaling/s1.1-32B

[4]Due to licensing and regulatory restrictions, we were unable to directly use certain models (e.g., GPT-4o).

*Table 1.* **Comparison of fine-tuning methods on LLaMA3.1-8B Instruct and S1.1-3B.** Results are averaged over math-focused (Group 1) and general reasoning (Group 2) benchmarks using 100K WebInstruct prompts. Teacher: LLaMA3.3-70B Instruct for CGD and CFT; CFT* uses GPT-4o data from Wang et al. (2025b). CGD achieves the strongest average performance across both model families. Bold denotes best, underline denotes second best. Δ rows report improvements of CGD over CFT. Cross-family results on Qwen2.5-Math-7B are shown in Table 3.

| Method | Math Reasoning Tasks (Group 1) | | | | | | General Reasoning Tasks (Group 2) | | | |
|---|---|---|---|---|---|---|---|---|---|---|
| | MATH500 | Minerva-Math | GSM8K | OlympiadBench | AMC23 | Avg. | TheoremQA | GPQA | MMLU-PRO | Avg. |
| LLaMA3.1-8B Instruct | 50.6 | 33.5 | 85.3 | 14.5 | 22.5 | 41.3 | 27.6 | 30.8 | 31.2 | 29.9 |
| + *SFT* | 41.2 | 24.6 | 80.7 | 10.8 | 20.0 | 35.5 | 22.1 | 33.3 | 39.3 | 31.6 |
| + *Distilled SFT* | 53.4 | 32.7 | 85.3 | 19.6 | 27.5 | 43.7 | 28.9 | 31.8 | 35.1 | 31.9 |
| + *CFT* with GPT-4o* | **54.8** | 33.1 | **86.2** | 18.2 | 25.0 | 43.5 | **35.0** | 30.3 | **40.8** | 36.4 |
| + *CFT* | 51.8 | 32.7 | 84.8 | 15.7 | 22.5 | 41.5 | 28.2 | 34.3 | 34.2 | 32.4 |
| + *CGD (ours)* | 54.2 | **33.6** | 85.7 | **23.7** | **37.5** | **46.9** | 34.0 | **35.9** | 40.3 | **36.7** |
| Δ = CGD - CFT | 2.4 | 0.9 | 0.9 | 8.0 | 15.0 | 5.4 | 5.8 | 1.6 | 6.1 | 4.3 |
| S1.1-3B | 54.0 | 16.9 | 76.8 | 20.6 | 30.0 | 35.4 | 21.6 | 16.7 | 13.7 | 17.9 |
| + *SFT* | 55.4 | 18.8 | 76.8 | 19.6 | 30.0 | 40.1 | 22.8 | 29.8 | **36.9** | 29.8 |
| + *Distilled SFT* | 60.6 | 22.1 | **83.1** | 20.4 | 22.5 | 41.7 | **34.9** | 29.3 | 36.4 | **33.5** |
| + *CFT* | 49.6 | 21.0 | 77.3 | 19.3 | 27.5 | 38.9 | 25.9 | 26.7 | 35.9 | 29.5 |
| + *CGD (ours)* | **61.8** | **27.9** | 82.5 | **23.1** | **35.0** | **46.1** | 32.8 | **31.8** | 35.7 | 33.4 |
| Δ = CGD - CFT | 12.2 | 6.9 | 5.2 | 3.8 | 7.5 | 7.2 | 5.6 | 5.0 | -0.2 | 3.5 |

*Table 2.* **Effect of different fine-tuning strategies on LLaMA3.1-8B Instruct across diverse benchmarks.** While CGD preserve or improve performance, CFT severely degrades general capabilities.

| Method | IFEval | MUSR | TruthfulQA | BBH | HumanEval |
|---|---|---|---|---|---|
| LLaMA3.1-8B Instruct | 76.9 | 37.8 | 54.0 | **48.3** | 59.7 |
| + *SFT* | 76.6 | 36.9 | 52.0 | 48.0 | 57.8 |
| + *Distilled SFT* | **77.5** | 39.0 | 53.9 | 47.0 | 58.7 |
| + *CFT* w/ GPT4o* | 55.6 | 35.0 | 53.5 | 44.2 | 60.3 |
| + *CGD (ours)* | 76.1 | **39.3** | **54.5** | 47.1 | **64.6** |

uated LLaMA3.1-8B Instruct models on HumanEval (Chen et al., 2021) (Python code generation) zero-shot as shown in Table 2. CGD achieves +4.9% Pass@1 improvement (59.7% → 64.6%) over LLaMA3.1-8B Instruct despite being trained on data without code, and outperforming CFT (+4.3%). This demonstrates CGD's self-correction transfers to out-of-distribution tasks, confirming it learns generalizable reasoning patterns.

### 4.3 Cross-Family Validation on Qwen2.5-7B-Math

To further validate the robustness and cross-family effectiveness of our approach, we apply CGD to the Qwen2.5-7B-Math model and compare against multiple strong baselines. We evaluate against: (1) the official Critique Fine-Tuning (CFT) checkpoint [5], trained with GPT-4o as teacher, and (2) our method with both a frontier teacher (Claude Sonnet 3.7) and a weaker open-source teacher (S1.1-32B).

The results in Table 3 demonstrate several key findings. First, CGD with Claude Sonnet 3.7 achieves the strongest overall performance (50.4 avg), outperforming CFT (48.9) and representing a **+22.6% gain over the base model** (27.8 → 50.4). Second, even when using the significantly weaker

[5] https://huggingface.co/TIGER-Lab/Qwen2.5-Math-7B-CFT

S1.1-32B teacher, CGD maintains competitive performance (49.0 avg, +21.2% over base), demonstrating teacher robustness. This demonstrates again with another model family that CGD can achieve state-of-the-art performance without relying on the most powerful closed-source models, highlighting its practical advantages for resource-constrained settings. Third, the consistent gains across architectures validate that CGD generalizes effectively across model families, scales, and teacher qualities: LLaMA3.1-8B (+15.0% on AMC23, +8.0% on OlympiadBench), S1.1-3B (+10.7% on math reasoning, +3.9% on general reasoning over base model), and Qwen2.5-Math-7B (+22.6% over base). Additional cross-architecture validation ablations on Mixtral-8x7B and OLMo-7B are provided in Appendix B.

Interestingly, S1.1-32B outperforms Claude Sonnet 3.7 on Minerva-Math (48.5 vs. 44.1), suggesting that teacher-student compatibility may matter more than raw teacher strength for domain-specific benchmarks. We hypothesize that S1.1's math-specialized pretraining produces critiques more aligned with Qwen's reasoning style.

### 4.4 The Role of the Critique as a Learning Signal

To isolate the impact of the critique as a learning signal during fine-tuning, we compare our CGD method against a key ablation variant, *CGD without Critique*. Both models see identical input prompts and refined answers. The only difference is the presence of the critique in the conditioning context during training.

This ablation also distinguishes CGD from explanation distillation methods like ORCA (Mukherjee et al., 2023). ORCA conditions training on generic teacher explanations of correct solutions; CGD conditions on critiques of the student's specific errors. Our "CGD without Critique" ablation, which receives identical refined answer targets but no error

*Table 3.* **Cross-family validation on Qwen2.5-Math-7B.** CGD achieves the strongest performance with both frontier (Claude Sonnet 3.7) and open-source (S1.1-32B) teachers, outperforming CFT. We additionally include a CFT baseline trained with the same S1.1-32B teacher to provide a fully controlled comparison.

| Method | Teacher Model | MATH500 | Minerva-Math | OlympiadBench | AMC23 | AIME24 | Avg. |
|---|---|---|---|---|---|---|---|
| Qwen2.5-Math-7B (Base) | - | 55.4 | 13.6 | 19.9 | 40.0 | 10.0 | 27.8 |
| CFT | GPT-4o | 79.2 | 45.2 | 40.7 | 62.5 | 16.7 | 48.9 |
| CFT | S1.1-32B | 71.6 | 27.9 | 36.6 | 55.0 | 20.0 | 46.9 |
| **CGD (Ours)** | Claude Sonnet 3.7 | 79.4 | 44.1 | 41.2 | **67.5** | **20.0** | **50.4** |
| **CGD (Ours)** | S1.1-32B | **79.6** | **48.5** | **41.3** | 62.5 | 13.3 | 49.0 |

diagnosis, is analogous to explanation distillation.

As shown in Figure 3, removing critique conditioning leads to substantial degradation (up to 114% degradation), demonstrates that student-specific error feedback provides supervision beyond what generic solution traces offer. Since critiques are **never present at inference** for either model, this gap indicates that critiques function as a training-only optimization scaffold, shaping how the model learns to correct errors rather than just mapping prompts to answers. By conditioning refinement on explicit error diagnoses during training, CGD enables the model to more effectively learn the transformation $f(\text{error} \rightarrow \text{correction})$, instead of directly optimizing $f(\text{prompt} \rightarrow \text{answer})$ under weaker supervision.

Beyond the binary presence of critiques, we ask whether the model conditions on critique *content* during training. We constructed a controlled ablation over $\sim$10K problems (a subset of our full 100K training set) with three conditions sharing identical student answers and refined-answer targets, varying *only* the critique text: (A) specific, relevant critiques identifying the student's exact error; (B) real critiques from unrelated problems (specific but irrelevant); and (C) a fixed generic string. Table 4 reports results averaged over 3 seeds.

*Table 4.* **Critique content ablation.** Only the critique text varies; all other training variables are held constant. Mean $\pm$ std over 3 seeds. Full details in Appendix I.

| | Critique Content | MATH-500 | AMC23 | AIME24 |
|---|---|---|---|---|
| A | Specific & relevant | **55.5±0.6** | **31.7±1.4** | **12.2±1.9** |
| B | Specific but irrelevant | 53.3±0.5 | 18.3±1.7 | 6.7±0.0 |
| C | Generic | 54.0±0.4 | 26.7±1.4 | 7.8±1.9 |

Specific, relevant critiques (A) outperform the generic baseline (C) on all reasoning-intensive benchmarks (+1.5 MATH-500, +5.0 AMC23, +4.4 AIME24), confirming that specificity matters. Crucially, irrelevant critiques (B) perform *worse* than even the generic placeholder on AMC23 and AIME24: when critique content systematically mismatches the problem, it actively misleads optimization. The ordering A > C > B demonstrates that the student reads and internalizes critique content during training rather than treating it as inert context. This training-time sensitivity to critique

quality is complementary to the inference-time robustness shown in Section 4.5: once trained on high-quality critiques, the model can discard adversarial noise at test time, but misleading critiques during training prevent effective learning in the first place.

### 4.5 Diagnostic Analysis

CGD decouples critique consumption (training) from critique generation (inference), requiring the model to compress external guidance into internal representations. We present three analyses characterizing this internalization.

**1. Internalization vs. Memorization.** We analyze whether CGD memorizes critique phrasing or learns underlying concepts by measuring lexical overlap between critiques and refined answers on 50K training examples (Appendix D.7). Token overlap is moderate (16.6%), reflecting shared mathematical terminology, but bigram overlap is low (5.7%). This gap indicates the model generates solutions using its own reasoning patterns rather than copying critique text. Supporting this, conditioning on critiques reduces gradient norms by 27% during training (Appendix D.1), consistent with critiques providing a more direct optimization path.

**2. Generalization to Hard Problems (AIME 2024).** We evaluate generalization on AIME 2024, a benchmark of challenging competition math problems outside the training distribution. Table 5 presents results under greedy decoding (Pass@1) and stochastic sampling (Pass@k with temperature 0.6, top-p 1.0; mean $\pm$ std over 3 seeds, with all other parameters held constant between models). Each Pass@k value reports the mean and standard deviation over three independent runs with different random seeds.

CGD improves greedy accuracy **5×** over the base model (3.3% $\rightarrow$ 16.7%) with **4.4×** longer reasoning chains (477 $\rightarrow$ 2110 words). Under stochastic sampling, CGD outperforms at Pass@8 (23.3% vs. 18.9%) and Pass@16 (34.4% vs. 26.7%), with gains strongest at low sample budgets, indicating improved reasoning quality rather than search. Full results appear in Appendix D.4.1; similar trends hold on AIME 2025 (Appendix D.4.2), indicating that CGD's gains on hard competition problems are robust.

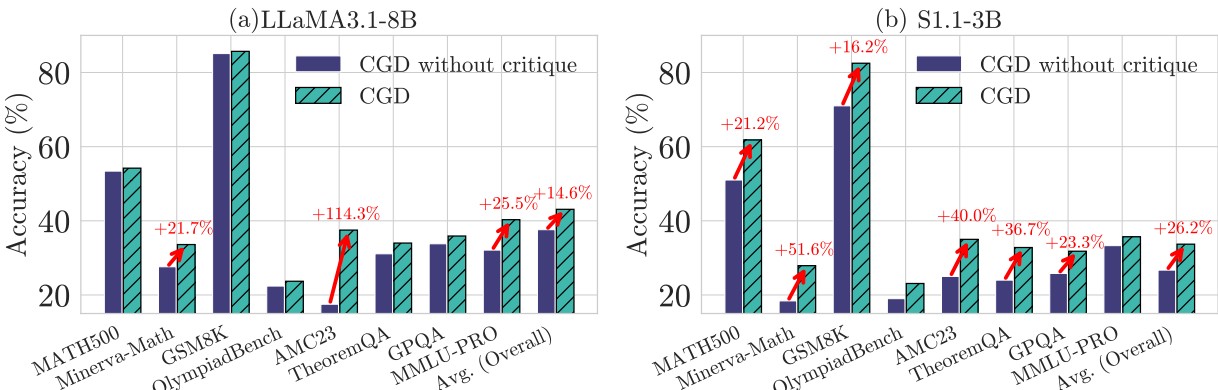

*Figure 3.* **Impact of critique conditioning.** Removing critiques from training significantly degrades performance, with relative drops of up to 114% on AMC23. This confirms critiques provide essential learning signal beyond input-output pairs alone.

*Table 5.* **Performance on AIME 2024.** Evaluated on LLaMA3.1-8B Instruct (teacher: LLaMA3.3-70B Instruct). CGD improves both greedy accuracy and sample efficiency. Pass@1 uses greedy decoding (temperature 0); Pass@k results report mean ± std over 3 seeds with temperature 0.6, top-p 1.0. CGD's gains are strongest at low k, indicating improved reasoning quality rather than search.

| Metric | Base Model | CGD |
|---|---|---|
| Pass@1 (greedy) | 3.3% | **16.7%** |
| Pass@8 | 18.9 ± 3.1% | **23.3 ± 0.0%** |
| Pass@16 | 26.7 ± 2.7% | **34.4 ± 4.2%** |
| Avg. Reasoning Length | 477 words | **2110 words** |
| Avg. Reasoning Steps | 16.4 | **49.5** |

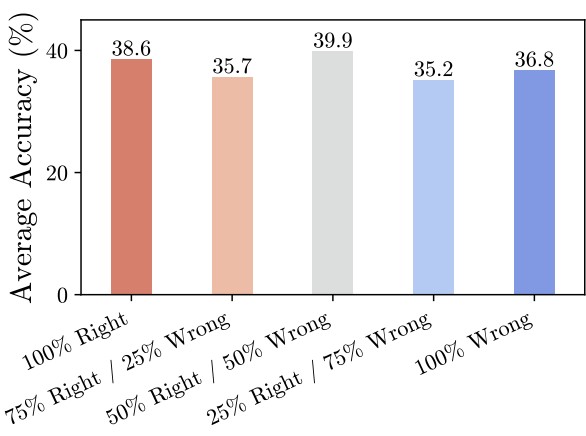

*Figure 4.* **Robustness via Data Diversity.** Performance peaks when the model is trained on a balanced 50/50 mixture of correct and incorrect student answers. This forces the model to distinguish between "reinforcement" and "correction" signals, preventing blind critique obedience.

## 3. Robustness to Critique Quality.

Does the model learn to reason or simply obey critiques? Our ablation on critique correctness (Figure 4) shows that a balanced 50/50 mixture of correct and incorrect student responses yields higher performance than training on corrections alone, indicating the model learns to evaluate critique validity against its own understanding, developing an internal discriminator that distinguishes informative from non-informative guidance.

Counterfactual case studies (Appendix D.5) on problems spanning algebra, number theory, and geometry confirm this: CGD ignores nonsensical adversarial critiques and solves problems correctly, while the base model is derailed. Combined with attention analysis showing high critique attention in early layers (Appendix D), these results indicate functional use of critiques rather than blind imitation.

An LLM-as-judge analysis of 20,000 training critiques (Appendix H) confirms that critiques for incorrect student answers exhibit significantly higher informativeness and specificity than those for correct answers, validating that our training data contains meaningful quality variation along both dimensions.

### 4.6 Training Stability

To ensure a rigorous comparison, we evaluated the learning rate sensitivity of CGD against the CFT baseline (Table 6). While CFT's performance degrades significantly at higher learning rates (dropping ≥ 9 points), CGD remains robust. This stability stems from CGD's objective: by training on the *refined answer* rather than the critique itself, CGD avoids the format drift inherent in critique-generation objectives like in CFT (see Appendix B.2.1 for more details).

*Table 6.* **Stability Analysis.** CGD outperforms CFT across learning rates, avoiding the instability caused by critique-generation format drift.

| Method | Avg Score (LR=1e-6) | Avg Score (LR=5e-6) |
|---|---|---|
| CFT | 39.3 | 30.1 (-9.2) |
| **CGD (Ours)** | **44.8** | **42.2** (-2.6) |

To contextualize CGD's computational requirements, Table 7 compares training costs with SimpleRL-Zero (Zeng et al., 2025), a recent RL-based reasoning method. We emphasize that this is **not an apples-to-apples comparison**: CGD and RL represent fundamentally different training paradigms. CGD achieves competitive aggregate performance at substantially lower compute. However, RL methods excel on competition-level problems (AIME24: 33.3% vs. 20.0%), likely because RL's exploration enables discovery of novel solution strategies beyond what teacher demonstrations provide. We view CGD and RL as complementary: CGD provides efficient supervised knowledge transfer, while RL provides exploration-based optimization. Whether combining them yields additional benefits remains an interesting direction for future work (see Appendix B.4 for additional discussion).

*Table 7.* **Compute efficiency context** (not a direct paradigm comparison). SimpleRL-Zero uses 1,152 H100 GPU-hours; CGD uses ∼48 A100 GPU-hours (8 training + ∼40 data generation).

| Base Model | Method | GPU Hours | Avg. |
|---|---|---|---|
| LLaMA3.1-8B | SimpleRL-Zero | 1152 (H100) | 10.6 |
| | CGD (Ours) | ∼48 (A100) | **11.9** |
| Qwen2.5-Math-7B | SimpleRL-Zero | 1152 (H100) | 48.9 |
| | CGD (Ours) | ∼48 (A100) | **50.4** |

## 5   Limitations

Our evaluation focuses on mathematical and structured reasoning, where errors admit clear diagnoses. Extending CGD to open-ended or creative domains, where correctness is subjective, may require alternative critique generation strategies.

CGD assumes access to informative critiques. Although we observe consistent improvements across teachers of varying quality, and our ablations show robustness to mixed critique quality (Appendix B.1.4), systematic study of failure modes under adversarial critiques remains future work.

CGD introduces data generation cost for teacher critiques and refined answers. While efficient at inference, this overhead may be nontrivial at scale. We view CGD as a cost-effective training paradigm complementary to existing approaches.

## 6   Conclusion

We introduced **C**ritique-**G**uided **D**istillation, a critique-conditioned fine-tuning framework that trains models to refine their own imperfect responses using explanatory feedback. By conditioning on critiques during training while requiring only the prompt at inference, CGD improves reasoning performance without architectural inference over-

head or output-format drift.

Across multiple model families and benchmarks, CGD consistently outperforms standard distillation and CFT, yielding substantial gains on challenging mathematical reasoning while preserving general capabilities such as instruction following. Notably, CGD exhibits consistent improvements across diverse model families, including LLaMA, Qwen, S1.1, Mixtral, and OLMo. Despite being trained on data containing no code, CGD transfers to out-of-distribution tasks such as code generation, highlighting its ability to learn generalizable reasoning patterns.

Our results position CGD as a practical and efficient training paradigm for reasoning tasks. Whether CGD and RL provide complementary benefits when combined remains an interesting direction for future work.

## Impact Statement

This paper presents work whose goal is to advance the field of Machine Learning. There are many potential societal consequences of our work, none which we feel must be specifically highlighted here.

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

## Supplementary Material

This supplementary material provides additional details, analyses, and resources for our work.

- **Appendix A: Experimental Setup and Hyperparameters.** Full details on training configuration, evaluation protocols, and hyperparameter settings.

- **Appendix B: Additional Experimental Results.** Extensive ablation studies including results on different student models, teacher models, math-specific training data (MetaMathQA), critique correctness mixtures, training curves, and compute efficiency context.

- **Appendix C: Theoretical Motivation.** Theoretical intuition connecting CGD to auxiliary task learning, curriculum learning, and analysis of why improvements persist at inference.

- **Appendix D: Detailed Diagnostic Analyses.** Full methodology and results for entropy analysis, gradient norm calculations, attention mechanism analysis, generalization to AIME 2024/2025, counterfactual robustness studies, and overlap analysis.

- **Appendix E: Critique and Refinement Generation Prompts.** Exact prompts used for generating teacher critiques and refined answers during data creation.

- **Appendix F: Example Training Data.** Representative CGD training samples and qualitative comparison of model responses.

- **Appendix G: Quantitative Criterion for Critique Receptivity.** The Relative Gradient Norm Reduction (RGNR) heuristic for predicting base model receptivity to CGD.

- **Appendix H: Critique Quality Analysis.** LLM-as-judge evaluation of 20,000 training critiques on informativeness and specificity.

- **Appendix I: Critique Content Ablation.** Controlled training ablation varying critique content (specific-relevant, specific-irrelevant, generic) to test sensitivity.

- **Appendix J: Code Instructions.** Summary of the codebase and instructions for reproducing training and evaluation results.

## A  Experimental Setup and Hyperparameters

### A.1  Experimental Setup

All experiments were conducted using NVIDIA A100 40GB GPUs. For training large-scale models, we employed DeepSpeed ZeRO-3 optimization for efficient memory and compute scaling across multiple GPUs, which enables optimizer state partitioning, gradient partitioning, and activation checkpointing to support training with larger batch sizes and model sizes.

We evaluate model performance using exact match accuracy, averaged over the test sets, and report mean performance over three random seeds to account for training variability.

### A.2  Hyperparameters

We provide the key hyperparameters used in training our models across all experiments. Unless otherwise noted, these values were held constant.

*Table 8.* **Summary of hyperparameters used in our experiments.**

| Hyperparameter | Value |
|---|---|
| Batch size | 64 |
| Learning rate | 1e-6 |
| Optimizer | AdamW |
| Scheduler type | cosine |
| Max sequence length | 8192 |
| Number of epochs | 1 |
| Warmup ratio | 0.1 |

## B  Additional Experimental Results

### B.1  Ablation Studies

In this section, we analyze the mechanisms behind CGD's effectiveness and study the impact of ablation studies. We demonstrate CGD's robustness across different training datasets and hyperparameters. We further analyze the impact of the training data's critique composition in Appendix B.1.4, finding that a balanced mixture of feedback yields the most robust model.

#### B.1.1  ABLATION: GENERALIZATION TO MATH-SPECIFIC TRAINING DATA

To test the generalizability of our method, we conducted experiments using MetaMathQA, a math-reasoning-focused dataset. As shown in Table 9, CGD again demonstrates strong performance, outperforming all baselines on on Group 1 and Group 2. This confirms that the benefits of the CGD framework are not limited to a specific data source. Notably, CGD surpasses the strongest baseline, i.e., CFT, on the advanced MATH500 and OlympiadBench challenges, yet shows slightly lower performance on Minerva-Math and GSM8K, which consist of middle-school to undergraduate-level problems.

*Table 9.* **Comparison of fine-tuning methods on LLaMA3.1-8B Instruct across math and reasoning tasks using 100K MetaMathQA examples with LLaMA3.3-70B Instruct as the teacher model.** Optimal results are highlighted in bold, while suboptimal outcomes are underlined. The Avg. columns represent the average performance across Groups 1 and 2, respectively.

| Method | Math Reasoning Tasks (Group 1) | | | | | | General Reasoning Tasks (Group 2) | | | |
| | MATH500 | Minerva-Math | GSM8K | OlympiadBench | AMC23 | Avg. | TheoremQA | GPQA | MMLU-PRO | Avg. |
|---|---|---|---|---|---|---|---|---|---|---|
| LLaMA3.1-8B Instruct | 50.6 | 33.5 | 85.3 | 14.5 | 22.5 | 41.3 | 27.6 | 30.8 | 31.2 | 29.9 |
| + *SFT* | 47.8 | 29.8 | 85.5 | 13.6 | 27.5 | 40.8 | 28.1 | **32.8** | 37.0 | 32.6 |
| + *Distilled SFT* | 50.2 | 33.5 | 79.8 | 18.5 | **35.0** | 43.4 | 31.2 | 28.8 | 28.1 | 29.4 |
| + *CFT* | 52.8 | **36.4** | **88.6** | 17.2 | 32.5 | 45.5 | 31.1 | 30.7 | **38.3** | 33.4 |
| + *CGD* | **59.0** | 34.6 | 87.3 | **21.8** | 32.5 | **47.0** | **34.1** | 30.3 | 36.1 | **33.5** |
| Δ = CGD - CFT | 6.2 | -2.2 | -1.3 | 4.6 | 0.0 | 1.5 | 3.0 | -0.5 | -2.2 | 0.1 |

### B.1.2 ABLATION: RESULTS ON DIFFERENT STUDENT MODELS

To assess the generality of our approach beyond the LLaMA model family, we replicate our main fine-tuning comparisons using Mixtral-8x7B Instruct v0.1 and OLMo-2-1124-7B-Instruct as the student models. Table 10 summarizes the results for both Mixtral-8x7B Instruct v0.1 and OLMo-2-1124-7B-Instruct student models across both math-focused and general reasoning benchmarks, with all models trained on the same 100K WebInstruct prompts.

Notably, we find that our method, CGD, consistently outperforms the baselines in both task groups for a different student model Mixtral-8x7B Instruct as shown in Table 10. On math reasoning tasks (Group 1), CGD achieves a +3.2% improvement over CFT. This includes substantial gains on GSM8K (+11.2%), Minerva-Math (+3.3%), and MATH500 (+3.4%), confirming transferability to a different architecture. In general reasoning tasks (Group 2), CGD shows a +1.3% average improvement over CFT, with notable gains on MMLU-PRO (+7.3%) and TheoremQA (+2.8%). While performance slightly declines on AMC23 (-2.5%) and GPQA (-6.0%) relative to CFT, these drops are not large enough to offset the overall performance improvements.

In contrast, CGD yields smaller gains on OLMo, i.e., 0.4 points less gain on Group 1 Avg. compared to Mixtral. While OLMo and Mixtral are similar in scale and baseline strength, they may differ in their ability to absorb critique-structured inputs. One possible explanation is differences in alignment data quality and fine-tuning objectives: prior work ( (Bai et al., 2022; Liang et al., 2025; Moon et al., 2025; Liu et al., 2024)) suggests that models tuned with richer dialogue-style data better leverage multi-step feedback. These results highlight that CGD is most effective when the student has been trained with supervision formats resembling critique/refinement, and they motivate deeper investigation into model-specific receptivity to critique-based training.

Importantly, CGD achieves consistently higher scores than SFT and CFT across most benchmarks, suggesting that distillation from critiques offers a more stable supervision signal than critique generation alone. These results gener-

alize our main findings and further support the modularity and versatility of our proposed training framework, highlighting that critique-based supervision is effective even for non-LLaMA models.

**Understanding Variation in Gains Across Model Families.** Our experiments reveal an important empirical pattern: CGD shows consistent improvements across model families, but the magnitude varies. Based on existing literature on instruction-tuning and data quality (Bai et al., 2022; Liang et al., 2025; Moon et al., 2025; Liu et al., 2024), we believe two factors best account for this variation: (A) alignment data quality and receptivity to critique-structured inputs, and (B) architectural and pretraining-induced inductive biases.

Prior work has shown that the quality and format of instruction-tuning data strongly affects downstream alignment and task performance (Bai et al., 2022; Liang et al., 2025). Models exposed to richer, more diverse, or interactive alignment data tend to make better use of supervision signals structured as dialogue, critique, or contrastive feedback. This is consistent with our empirical pattern:

- **LLaMA3.1-8B** (extensively instruction-tuned): large gains (+15.0% AMC23, +8.0% OlympiadBench, +5.4% math avg over CFT)

- **S1.1-3B** (math-specialized with strong reasoning priors): large gains (+10.7% math reasoning avg over base, +7.2% over CFT)

- **Qwen2.5-Math-7B** (math-specialized with strong reasoning priors): large gains (+22.6% over base, 27.8% → 50.4% avg)

- **Mixtral-8x7B** (MoE): modest gains (+3.2% avg over CFT)

- **OLMo-7B** (different pretraining corpus): modest gains (+2.8% avg over CFT)

These correlations suggest that a model's prior alignment to supervision and its baseline reasoning ability can amplify the benefit of CGD. Given these differences in pretraining corpora, alignment objectives, and architectural inductive

*Table 10.* **Evaluation of fine-tuning methods on Mixtral-8x7B Instruct across math-focused (Group 1) and general reasoning (Group 2) benchmarks, using WebInstruct as the training set.** CGD achieves the strongest performance in both groups, despite Mixtral being a different architecture than LLaMA. All methods are fine-tuned on 100K WebInstruct samples. Bold numbers denote the best, and underlined values indicate the second-best performance. The $\Delta$ row shows CGD's gains over the CFT baseline.

| Method | Math Reasoning Tasks (Group 1) | | | | | | General Reasoning Tasks (Group 2) | | | |
|---|---|---|---|---|---|---|---|---|---|---|
| | MATH500 | Minerva-Math | GSM8K | OlympiadBench | AMC23 | Avg. | TheoremQA | GPQA | MMLU-PRO | Avg. |
| Mixtral-8x7B Instruct | 29.6 | 15.4 | 69.4 | 8.9 | 7.5 | 26.2 | 21.2 | 21.7 | 24.7 | 22.5 |
| + *SFT* | 31.4 | 15.6 | 65.6 | 7.9 | 5.0 | 25.1 | 20.4 | 20.5 | **25.2** | 22.0 |
| + *CFT* | 35.6 | 20.6 | 63.8 | 11.1 | **10.0** | 28.2 | 23.6 | **31.8** | 16.0 | 23.8 |
| + *CGD* | **39.0** | **23.9** | **75.0** | **11.7** | 7.5 | **31.4** | **26.4** | 25.8 | 23.3 | **25.1** |
| $\Delta$ = CGD - CFT | 3.4 | 3.3 | 11.2 | 0.8 | -2.5 | 3.2 | 2.8 | -6.0 | 7.3 | 1.3 |
| OLMo-2-1124-7B-Instruct | 35.4 | 16.5 | 81.9 | 11.0 | 7.5 | 30.5 | 23.0 | 28.3 | 34.1 | **28.5** |
| + *SFT* | 36.4 | 15.1 | 80.5 | 11.0 | 12.5 | 31.1 | 19.1 | 28.1 | 34.1 | 27.2 |
| + *CFT* | 35.9 | 16.8 | 81.2 | 11.8 | 10.0 | 31.1 | 19.3 | 27.4 | 33.4 | 26.7 |
| + *CGD* | **37.4** | **16.9** | **83.2** | **12.1** | **20.0** | **33.9** | **24.2** | **28.3** | **34.2** | 28.2 |
| $\Delta$ = CGD - CFT | 1.5 | 0.1 | 2.0 | 0.3 | 10.0 | 2.8 | 4.9 | -1.1 | 0.7 | 1.5 |

biases, variation in the magnitude of CGD's gains is expected. We formalize this observation as the Relative Gradient Norm Reduction (RGNR) heuristic in Appendix G, which provides a practical diagnostic for predicting model receptivity to critique-guided training. While our current results support this interpretation, future work needs to perform more controlled experiments to gain deeper insights into this important phenomenon.

### B.1.3 ABLATION: RESULTS USING DIFFERENT TEACHER MODELS

We find that CGD provides consistent improvements over the base LLaMA3.1-8B Instruct model across both math and general reasoning benchmarks, regardless of the choice of teacher model as shown in Table 11. Using LLaMA3.3-70B Instruct as the teacher yields strong gains, particularly in general reasoning tasks, while adopting the open-weight S1.1-32B teacher leads to even stronger performance on several challenging math benchmarks. For example, CGD with S1.1 improves AMC23 accuracy by +20.0 absolute points (22.5 ⇒ 42.5). These results suggest that the benefits of CGD are not limited to teacher scale or architecture family; even when transferring critiques from a non-LLaMA teacher, the student acquires improved reasoning ability.

We emphasize that the teacher ablation in Table 11 holds the student fixed (LLaMA3.1-8B) while varying the teacher model. In contrast, the S1.1-3B results presented in Table 1 focus on the student-side generalization, where the model itself is smaller and trained with critiques and responses from S1.1-32B.

Importantly, these findings support the claim that CGD's effectiveness is not solely determined by the raw strength of the teacher, but also by the structured way in which critiques are generated and incorporated during training. While stronger teachers such as GPT-4o or future generations of S1.1 may offer further improvements, our preliminary experiments already demonstrate that critique quality

and integration play a critical role in driving gains. In other words, CGD does more than transfer answers, i.e., it teaches the student how to reason through structured critique, enabling performance improvements that extend beyond what is achievable with standard distillation.

### B.1.4 ABLATION: IMPACT OF CRITIQUE CORRECTNESS MIXTURE

This section provides extended methodology for the critique mixture ablation presented in Figure 4. We trained five models on WebInstruct with varying ratios of correct/incorrect student answers (as indicated by critique conclusions). Total sample size (25K) and hyperparameters were held constant. Ratios tested: 100% correct, 75/25, 50/50, 25/75, and 100% incorrect.

Figure 4 shows that **average** performance across benchmarks peaks at 50/50 mixture (39.9%), outperforming both extremes (100% correct: 38.6%, 100% incorrect: 36.8%). The key finding is not that 50/50 is universally optimal, but that **neither extreme is sufficient**: training on corrections-only (100% incorrect) prevents learning what good reasoning looks like, while training on reinforcement-only (100% correct) prevents learning to distinguish valid from invalid reasoning. Balanced exposure develops an internal discriminator, the same capability that enables robustness to adversarial critiques (Appendix D.5). We leave systematic study of optimal mixture ratios across domains to future work.

### B.2 Epoch-Accuracy Curves

Figure 5 shows the progression of final accuracy across training epochs for **C**ritique-**G**uided **D**istillation on six math-focused benchmarks. We observe that performance is generally stable throughout training, with no substantial drops in accuracy for any dataset. While the upward trends are not particularly pronounced, the lack of degradation suggests that our method is robust to overfitting and avoids catastrophic forgetting. In particular, benchmarks such as

*Table 11.* **Comparison of CGD using different teacher models on the student model LLaMA3.1-8B Instruct across math (Group 1) and general reasoning (Group 2) benchmarks, using WebInstruct as the training set.** Using S1.1 as the teacher model achieves a stronger performance in complex math-reasoning tasks, despite S1.1 being a different architecture than LLaMA.

| Method | Math Reasoning Tasks (Group 1) | | | | | | General Reasoning Tasks (Group 2) | | | |
|---|---|---|---|---|---|---|---|---|---|---|
| | MATH500 | Minerva-Math | GSM8K | OlympiadBench | AMC23 | Avg. | TheoremQA | GPQA | MMLU-PRO | Avg. |
| Initialized from LLaMA3.1-8B Instruct | | | | | | | | | | |
| LLaMA3.1-8B Instruct | 50.6 | 33.5 | 85.3 | 14.5 | 22.5 | 41.3 | 27.6 | 30.8 | 31.2 | 29.9 |
| + *CGD with LLaMA3.3-70B* | 54.2 | 33.6 | 85.7 | 23.7 | 37.5 | 46.9 | 34.0 | 35.9 | 40.3 | 36.7 |
| + *CGD with S1.1-32B* | 56.8 | 37.1 | 86.8 | 16.7 | 42.5 | 48.0 | 32.2 | 34.3 | 40.4 | 35.7 |
| Teacher Models | | | | | | | | | | |
| LLaMA3.3-70B Instruct | 75.3 | 55.9 | 96.1 | 39.3 | 65.0 | 66.3 | 53.6 | 37.9 | 70.6 | 54.0 |
| S1.1-32B | 92.9 | 58.1 | 94.8 | 63.6 | 85.0 | 78.9 | 64.4 | 46.0 | 48.3 | 52.9 |

MATH (increases from 55.8 to 56.7) and OlympiadBench (increases from 22 to 23.3) show modest improvements, indicating some continued learning over time. These curves offer cautious empirical support for the consistency and stability of our fine-tuning process.

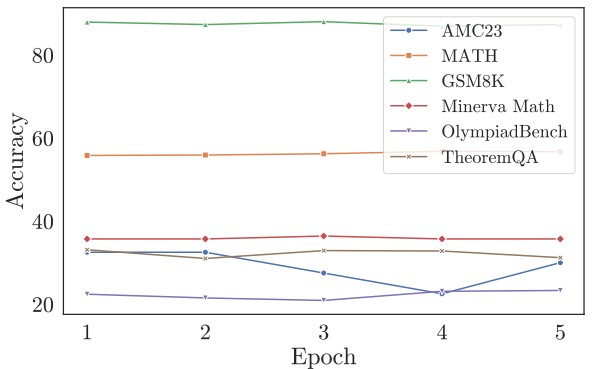

*Figure 5.* **Accuracy over training epochs for CGD on six math-focused benchmarks.** While trends are modest, performance remains stable throughout, indicating resistance to overfitting and catastrophic forgetting.

### B.2.1 TRAINING STABILITY AND HYPERPARAMETER ROBUSTNESS

To ensure a rigorous and fair comparison, we evaluate the learning rate sensitivity of CGD against the CFT baseline using the same teacher model (LLaMA3.3-70B Instruct) and identical prompts from the WebInstruct subset (Table 12). Both methods are trained on 100K critique-augmented examples under identical training schedules, varying only the learning rate between $1 \times 10^{-6}$ and $5 \times 10^{-6}$. While CFT's performance significantly degrades at the higher learning rate, dropping by over 9 points on average, CGD remains robust and outperforms CFT across all metrics regardless of learning rate. These results suggest that CGD's structured self-correction task with the use of both critiques and refined answers enables more stable optimization and better generalization, even under suboptimal hyperparameter choices, whereas CFT remains brittle to training dynamics despite

access to the same supervision signals.

Figure 6 depicts how both methods respond to changes in learning rate. Figures (a) and (b) show the accuracy vs. learning-rate curves for our approach and CFT, respectively. Our method exhibits a smooth decline as the learning rate increases (Fig. 6a), whereas CFT's performance degrades more sharply (Fig. 6b).

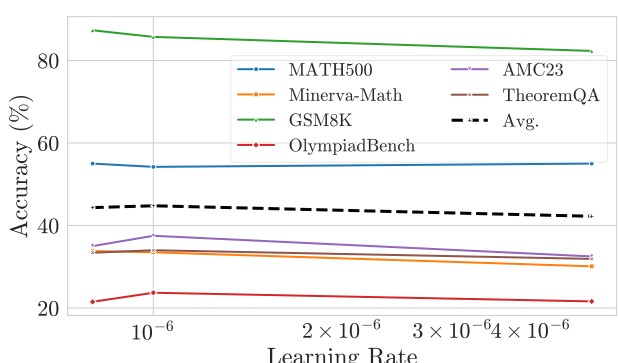

*(a)* CGD

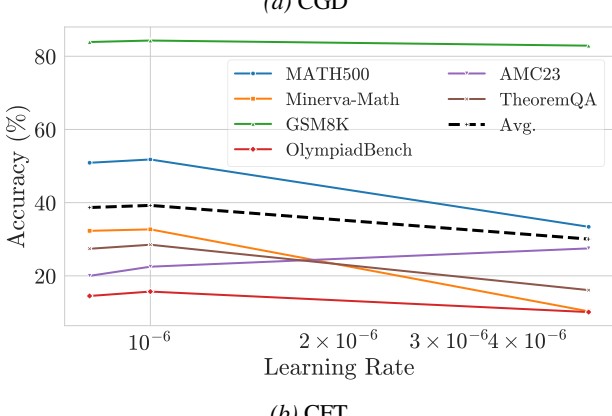

*(b)* CFT

*Figure 6.* **Accuracy vs. learning rate for (a) CGD (our method) and (b) the CFT baseline across six benchmarks.**

*Table 12.* **Comparison of CGD and CFT.** CGD consistently outperforms CFT across all benchmarks and is relatively robust to learning rate changes, while CFT exhibits significant performance degradation at higher learning rates. The Avg. column reflects average performance across all tasks.

| Method | MATH500 | Minerva-Math | GSM8K | OlympiadBench | AMC23 | TheoremQA | Avg. |
|---|---|---|---|---|---|---|---|
| CFT (LR = $1 \times 10^{-6}$) | 51.8 | 32.7 | 84.8 | 15.7 | 22.5 | 28.5 | 39.3 |
| CFT (LR = $5 \times 10^{-6}$) | 33.4 | 10.3 | 82.9 | 10.1 | 27.5 | 16.1 | 30.1 |
| CGD (LR = $1 \times 10^{-6}$) | 54.2 | 33.6 | 85.7 | 23.7 | 37.5 | 34.0 | 44.8 |
| CGD (LR = $5 \times 10^{-6}$) | 55.0 | 30.1 | 82.3 | 21.6 | 32.5 | 31.9 | 42.2 |

### B.3 Training Loss Analysis

We present training loss curves comparing **C**ritique-**G**uided **D**istillation and Critique-Finetuning (CFT) methods in Figure 7. The x-axis represents normalized training progress (%), and the y-axis shows the training loss.

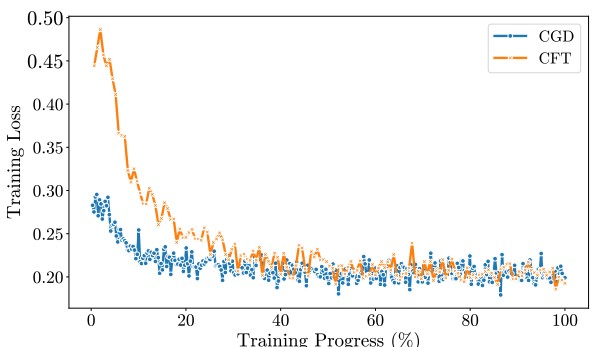

*Figure 7.* **Training loss comparison between CGD and CFT.** The x-axis indicates normalized training progress, and the y-axis shows loss.

From the plot, the CFT curve exhibits a noticeable initial spike in loss, which can be attributed to a format drift during early training. This is due to the model being trained on a critique-style instruction following dataset immediately after pretraining or SFT tuned on QA-style instructions. The shift from generating answers to critiquing Q&A pairs likely introduces a mismatch in expected input-output format, temporarily destabilizing the loss. As training progresses, however, the model adapts, and the loss curve stabilizes and declines.

In contrast, the CGD method shows a more stable and smooth decrease in loss throughout training, suggesting a more consistent and format-aligned supervision signal. This supports the hypothesis that CGD, by leveraging structured critiques without drastic task shifts, offers a gentler optimization trajectory and better alignment with initial model capabilities.

### B.4 Compute Efficiency Context

To contextualize CGD's computational requirements, we compare training costs with SimpleRL-Zero (Zeng et al., 2025), a recent RL-based reasoning method. We empha-size that **this is not an apples-to-apples comparison** since CGD and RL represent fundamentally different training paradigms with different objectives and tradeoffs:

We report the official numbers released by the SimpleRL [6] and compare them with our results on both LLaMA3.1-8B and Qwen2.5-7B-Math base models. Table 13 reports benchmark performance alongside compute costs. CGD achieves competitive aggregate performance at substantially lower compute. However, RL methods excel on competition-level problems (AIME24: 33.3% vs 20.0%), likely because RL's exploration enables discovery of novel solution strategies beyond what teacher demonstrations provide. CGD's strength lies in efficient knowledge transfer.

We view CGD and RL as complementary rather than competing approaches. CGD provides efficient supervised knowledge transfer from teachers; RL provides exploration-based optimization that can discover novel solutions. Whether combining CGD and RL yields additional benefits remains an interesting direction for future work.

## C Theoretical Motivation

This section provides theoretical intuition for why Critique-Guided Distillation (CGD) improves over standard distillation. Our goal is not to establish formal proofs, but to connect CGD's design to established learning principles in representation learning, auxiliary supervision, and curriculum learning. All arguments below should be interpreted as explanatory rather than predictive.

### C.1 CGD as Auxiliary Task Learning

CGD can be viewed through the lens of auxiliary task learning (Caruana, 1997; Ruder, 2017). During training, the model learns the mapping

$$f_\theta : (x, y', c) \mapsto \hat{y}, \tag{2}$$

where $x$ is the prompt, $y'$ is the student's initial response, and $c$ is a teacher-provided critique. At inference time, only $x$ is available. A central question is why training with auxiliary inputs $(y', c)$ improves performance when these inputs are absent at test time.

---

[6] https://github.com/hkust-nlp/simpleRL-reason

*Table 13.* **Compute efficiency context** (not a direct paradigm comparison). CGD and RL have fundamentally different objectives. We report these numbers to contextualize CGD's efficiency, not to claim superiority over RL methods. SimpleRL-Zero requires 1,152 H100 GPU-hours; CGD requires approximately 48 total A100 GPU-hours (8 training + $\sim$40 data generation).

| Model | Method | Data Size | GPU Hours | MATH500 | Minerva | OlympiadBench | AMC23 | AIME24 | Avg. |
|---|---|---|---|---|---|---|---|---|---|
| LLaMA3.1-8B | SimpleRL-Zero | 8K×12 | 1152 (H100) | 23.0 | 9.6 | 5.3 | 15.0 | 0.0 | 10.6 |
| | CGD (Ours) | 50K | $\sim$48 (A100) | 29.4 | 12.9 | 7.0 | 10.0 | 0.0 | **11.9** |
| Qwen2.5-Math-7B | SimpleRL-Zero | 8K×12 | 1152 (H100) | 77.2 | 33.5 | 37.9 | 62.5 | **33.3** | 48.9 |
| | CGD (Ours) | 50K | $\sim$48 (A100) | 79.4 | 44.1 | 41.2 | **67.5** | 20.0 | **50.4** |

**Representation Learning View.** Let $h_\theta(x)$ denote the internal representation of the prompt. Standard distillation trains representations that are sufficient for mapping $x$ to $\hat{y}$. In contrast, CGD trains representations that support correcting wrong reasoning when guided by critique information.

The critique $c$ provides a partial and task-specific signal indicating which aspects of the problem were misunderstood. To generalize across diverse critiques and student errors, the model is encouraged to learn representations of $x$ that are predictive of critique-relevant features, rather than relying on correlations between prompts and answers. In this sense, critiques act as structured auxiliary supervision that shapes $h_\theta(x)$ toward error-sensitive representations. Training to predict $\hat{y}$ from $(x, y', c)$ encourages alignment between $h_\theta(x)$ and the information highlighted by critiques, even when the critique itself is not available at inference.

**Optimization Signal Consistency.** Compared to standard distillation, CGD provides a more targeted training signal. In distillation, the model must infer which aspects of $x$ matter for producing $\hat{y}$ with limited guidance. In CGD, the critique explicitly localizes reasoning failures, narrowing the effective search space during optimization.

This suggests a more consistent optimization signal during training. Consistent with this intuition, we observe reduced variability in gradient norms during CGD training relative to distillation baselines, indicating more stable learning dynamics (Appendix D.1). We emphasize that this observation is empirical and serves as supporting evidence rather than a formal claim.

## C.2 Connection to Curriculum Learning

CGD naturally implements curriculum learning over error patterns through critique diversity. Training data includes both cases where the student's initial response is correct and cases where it is incorrect:

- **Positive critiques**, where the student response is largely correct, reinforce effective reasoning patterns.

- **Negative critiques**, where the student response is wrong, expose common error modes and their corrections.

Although critique polarity is not a perfect proxy for prob-

lem difficulty, it correlates with whether the model must revise or reinforce its reasoning. Our mixture ablation study (Figure 4) shows that balanced exposure to both critique types yields optimal performance. This aligns with curriculum learning theory (Bengio et al., 2009), which suggests that learning benefits from structured exposure to examples spanning varying levels of difficulty and feedback.

## C.3 Why Improvements Persist at Inference

A key question is why performance gains persist when critiques and initial responses are absent at inference. We hypothesize two complementary mechanisms.

**Internalized Self-Correction.** By repeatedly training on transformations from erroneous reasoning to corrected solutions, the model learns an implicit self-correction behavior. At inference, this manifests as more deliberate reasoning processes. Empirically, CGD-trained models produce substantially longer reasoning traces on unseen hard problems such as AIME 2024, with a 4.4× increase in average reasoning length (Table 5), alongside improved accuracy. This suggests that CGD encourages more thorough internal search rather than surface-level pattern matching.

**Representation Transfer.** The representations learned during CGD training remain useful even without explicit critique inputs. Through exposure to critique-conditioned correction, the model learns to attend to problem features that commonly give rise to errors. Attention analyses provide indirect evidence for this effect: CGD-trained models allocate more attention to problem structure in early layers compared to baselines, a pattern that is correlated with improved inference-time behavior (Appendix D).

## C.4 Limitations of This Analysis

This section offers conceptual grounding rather than formal guarantees. Several open questions remain:

- Quantifying the gap between training with critiques and inference without critiques.

- Characterizing which classes of problems benefit most from critique-conditioned training.

- Understanding failure modes when critiques are mislead-

ing or systematically biased.

Addressing these questions represents an important direction for future theoretical work.

# D    Detailed Diagnostic Analyses

This section provides the detailed methodology, full quantitative results, and visual analyses for the diagnostic experiments summarized in the main paper. All diagnostic experiments were conducted on a set of 500 samples randomly drawn from the OpenMathInstruct-2 dataset using LLaMA3.1-8B Instruct student model with the same hyperparameters.

## D.1    Diagnostic Experiments

**Entropy Calculation.** To measure predictive confidence, we performed a forward pass for each model on the diagnostic dataset, using the full '(Prompt + Student Answer + Critique)' context. This context is formatted using the model's specific chat template. We then isolated the model's logits for the single, next token that would begin the 'Refined Answer'. These logits were converted to a probability distribution via the softmax function, and the Shannon entropy $(H(X) = -\sum p(x) \log p(x))$ was calculated. A lower entropy value indicates higher confidence in the prediction.

**Gradient Norm Calculation.** To measure learning signal efficiency, we took each final trained model and performed a single forward and backward pass on a diagnostic sample to compute the cross-entropy loss against the target answer. We then calculated the total L2 norm of the full parameter gradient vector. This was done for two input conditions: one 'With Critique' and one 'Without Critique', allowing for a controlled analysis of the critique's impact on the update signal. Analysis of Table 14 reinforces the fact that conditioned on an informative critique the model is able to better predict the final response. We report mean gradient norms with standard error: with critique $1802.7 \pm 78.3$, without critique $2446.9 \pm 91.2$. The difference is statistically significant (paired $t$-test, $p < 0.001$). This reduces the loss and in turn the magnitude of the gradient norm. We observed this trend during the entire period as well.

*Table 14.* **Gradient norm analysis for the final trained CGD model.** The presence of a critique provides a more efficient signal, reducing the update magnitude by 27%.

| CGD Model Condition | Mean Gradient Norm | Std. Dev. Gradient Norm |
|---|---|---|
| Without Critique | 2446.9 | 2011.9 |
| With Critique | 1802.7 | 1765.5 |

**Attention Analysis.** To analyze the model's internal reasoning, we generated answers with maximum 8192 tokens

for each sample and collected the attention matrices from all 32 layers. These scores were then aggregated by averaging across all attention heads and normalized to represent the percentage of attention paid by each generated token to three distinct sections of the prompt: the 'Problem', the 'Student Answer', and the 'Critique'.

## D.2    Quantitative Analysis of Model Confidence

The behavioral differences observed in our case study are supported by our quantitative diagnostics. As shown in Table 15, the key finding is that the CGD model is statistically significantly more confident (lower entropy) than all other generative baselines on the self-correction task. The statistical significance of this result ($p < 10^{-4}$ vs. Distilled SFT) confirms that the CGD training process forges a uniquely robust and decisive reasoning agent.

*Table 15.* **Summary of predictive confidence (Mean Entropy), averaged over 500 samples from OpenMathInstruct 2.** Lower entropy is better. Significance markers (*, **, ***) denote the p-value of a paired t-test comparing each baseline to our CGD model.

| Model | Mean Entropy |
|---|---|
| Baseline SFT | 6.56*** |
| SFT | 6.62*** |
| Distilled SFT | 6.49*** |
| **CGD** | 6.44 |

*Significance:* *** $p < 0.001$

## D.3    Attention Mechanism Analysis

To provide a deeper mechanistic view of the CGD model's reasoning process, we analyzed its internal attention patterns, averaged over 50 samples from the OpenMathInstruct 2 dataset. For each sample, we generated up to 8192 new tokens, allowing the model to complete its reasoning naturally. Our key finding is that the model employs a sophisticated, multi-phase reasoning strategy, using the critique as a foundational signal that is internalized early and acted upon during generation. This is illustrated across three complementary visualizations.

Figure 8 presents the model's attention flow across different layers during the generation of an answer. Starting at the first layer and all the way to the middle layers, there is significant attention on both the critique and the student response. This shows that the model has learned to exploit the signals in an informative critique and the noisy student response (e.g., with attention to the Critique at 48.1% and the Student Answer at 36.0% at the very first generation step). In the later layers of the model, the primary focus is on getting the correct response and hence most of the

attention is on the problem.

Figure 9 confirms how different pieces of information are processed at different levels of abstraction. The plot shows the average attention paid to each prompt section across all 32 layers. The results show that direct attention to the Critique's raw tokens peaks at the very input (31.9% at Layer 0), suggesting a strong initial intake of the signal. The model's focus then shifts to the Student Answer, with attention peaking in the semantic middle layers (22.1% at Layer 13), precisely where attention to the critique also sees a secondary rise. This could be attributed to the fact that the model's most abstract reasoning, understanding the flaw and synthesizing the correction, happens in the middle of the network. Finally, attention to the Problem details consolidates and peaks in the late layers (94.8% at Layer 25) as the model formulates its final output.

Finally, Figure 10 provides a high-level summary of attention from different layers, broken down by generation phase, which reinforces these findings. The heatmaps for later, more semantic layers (16 and 31) visualize the "plan-then-execute" pattern, showing that the initial generation phases are dominated by attention to the critique (48.1% for Layer 31 at token 1). This is consistent with a model that has learned to use the critique as a foundational guide to initiate and structure its reasoning process. These observed attention patterns suggest that the CGD has acquired a sophisticated reasoning process: it internalizes the critique's guidance at an early stage and then acts upon this internalized knowledge in its final, semantic layers to plan and execute a corrected solution. The following section provides a direct behavioral test of this hypothesis.

## D.4 Generalization to Hard Problems

### D.4.1 GENERALIZATION TO AIME 2024

This section provides extended analysis of CGD's performance on AIME 2024, supporting the results in Table 5. We evaluated CGD-trained LLaMA3.1-8B Instruct against the base model on all 30 AIME 2024 problems. For greedy decoding (Pass@1), we use temperature 0. For stochastic sampling (Pass@k), we use temperature 0.6 and top-p 1.0, with all decoding parameters held constant between models. Each Pass@k value is computed as the mean over three independent runs with different random seeds; we report both mean and standard deviation.

Table 16 presents complete Pass@k results including Pass@32. CGD's advantage is most pronounced at low sample budgets (Pass@1: +406%, Pass@8: +23%, Pass@16: +29%) and narrows at high k (Pass@32: +3%). This pattern is consistent with CGD improving underlying reasoning quality, rather than relying primarily on brute-force search, thus, fewer samples are needed to find correct solutions.

At very high k, brute-force sampling allows even weaker models to eventually find solutions, equalizing performance.

*Table 16.* **Full Pass@k results on AIME 2024.** CGD's gains are strongest at low k, indicating improved reasoning quality rather than search. Results report mean ± std over 3 seeds (temperature 0.6, top-p 1.0).

| Metric | Base Model | CGD | Relative Gain |
|---|---|---|---|
| Pass@1 (greedy) | 3.3% | 16.7% | **+406%** |
| Pass@8 | 18.9 ± 3.1% | 23.3 ± 0.0% | +23% |
| Pass@16 | 26.7 ± 2.7% | 34.4 ± 4.2% | +29% |
| Pass@32 | 36.7 ± 2.7% | 37.8 ± 3.1% | +3% |
| Avg. Reasoning Length | 477 words | 2110 words | **4.4×** |
| Avg. Reasoning Steps | 16.4 | 49.5 | **3.0×** |

As k increases, performance differences naturally diminish, as extensive sampling allows even weaker models to occasionally recover correct solutions. In contrast, CGD exhibits its largest gains at Pass@1 and low k, indicating that its benefits primarily comes from improved quality of individual reasoning attempts and reduced dependence on extensive sampling. This is consistent with our other findings: CGD produces longer, more deliberate reasoning chains (4.4× longer) that systematically work through problems rather than guessing.

### D.4.2 GENERALIZATION TO AIME 2025

We additionally evaluate CGD on AIME 2025, which is widely regarded as more challenging and noisy than AIME 2024. As with AIME 2024, we evaluate both the base LLaMA3.1-8B Instruct model and the CGD-trained model under stochastic sampling with temperature 0.6 and top-p 1.0. Each Pass@k value reports the mean and standard deviation over three independent runs with different random seeds.

Table 17 shows that CGD consistently outperforms the base model across all evaluated k. While absolute accuracies remain low for both models due to the difficulty of AIME 2025, CGD yields substantial relative improvements at low and moderate sample budgets (Pass@8 and Pass@16). As in AIME 2024, the performance gap narrows at higher k, consistent with the expectation that extensive sampling allows weaker models to eventually find correct solutions.

*Table 17.* **Full Pass@k results on AIME 2025.** Results report mean ± std over 3 seeds (temperature 0.6, top-p 1.0). Absolute accuracies are low due to benchmark difficulty, but CGD consistently outperforms the base model.

| Metric | Base Model | CGD | Relative Gain |
|---|---|---|---|
| Pass@8 | 1.1 ± 1.6% | 5.6 ± 3.1% | +409% |
| Pass@16 | 4.4 ± 4.2% | 15.6 ± 11.0% | +255% |
| Pass@32 | 12.2 ± 8.3% | 20.0 ± 0.0% | +64% |

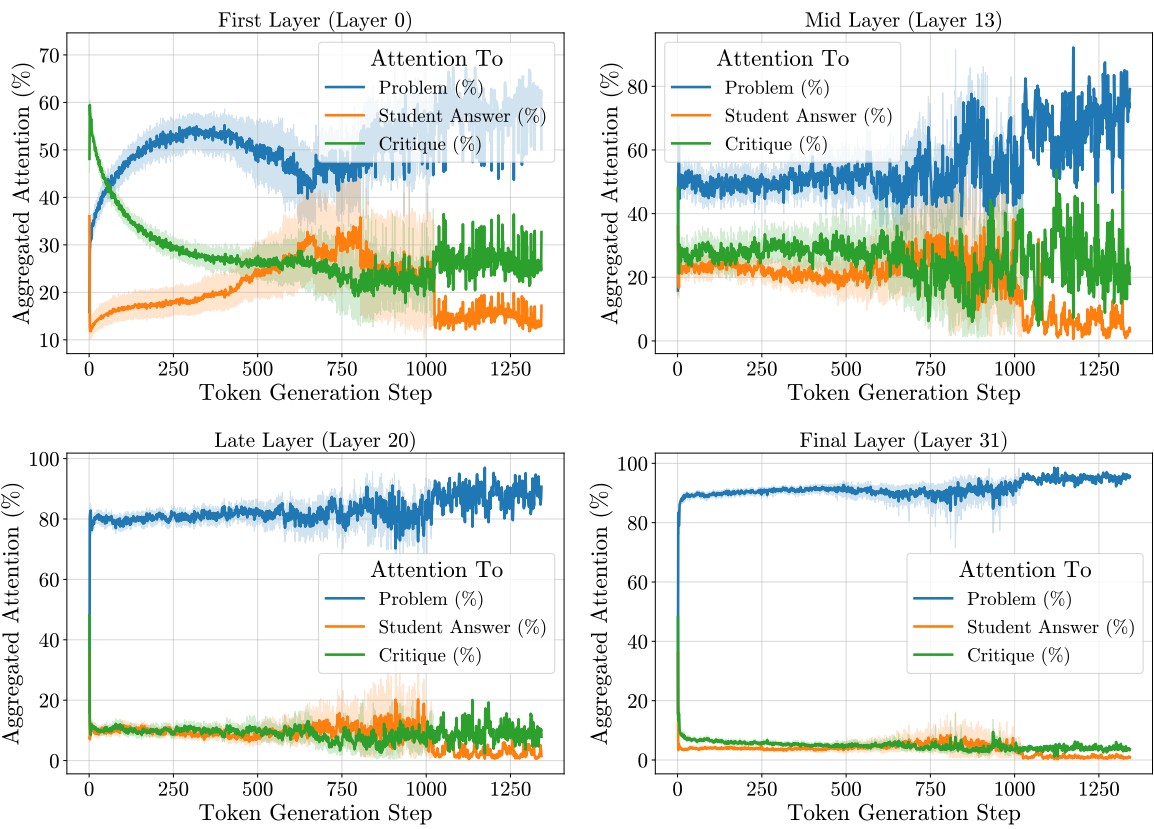

*Figure 8.* **Average attention flow of the CGD model.** All layers shown begin with a "planning" step, focusing on the Critique (48.1%) and Student Answer (36.0%). The final layer (bottom right) then pivots sharply to an "execution" phase, focusing on the Problem (> 90%), while the first layer (top left) continues to process the Critique. Shaded regions represent the 95% confidence interval over 50 samples.

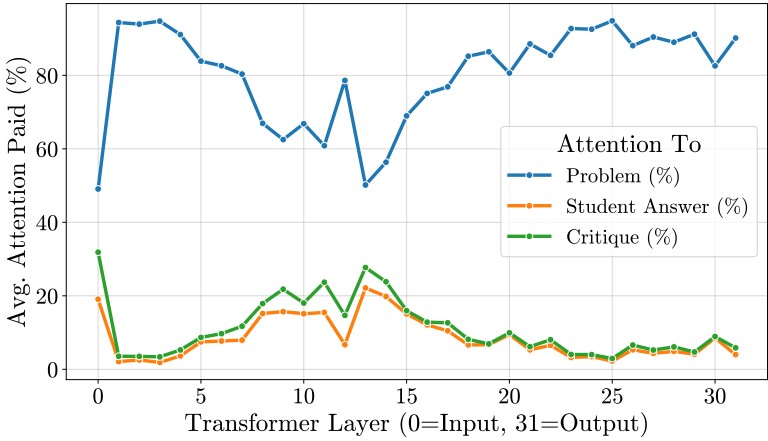

*Figure 9.* **Average attention paid to each prompt section across all 32 transformer layers.** The patterns suggest an *early intake* of the Critique (peak at Layer 0), followed by a *deep processing* of the Student Answer in conjunction with the critique in the semantic middle layers (peak at Layer 13). Attention to the **Problem** dominates in the final layers.

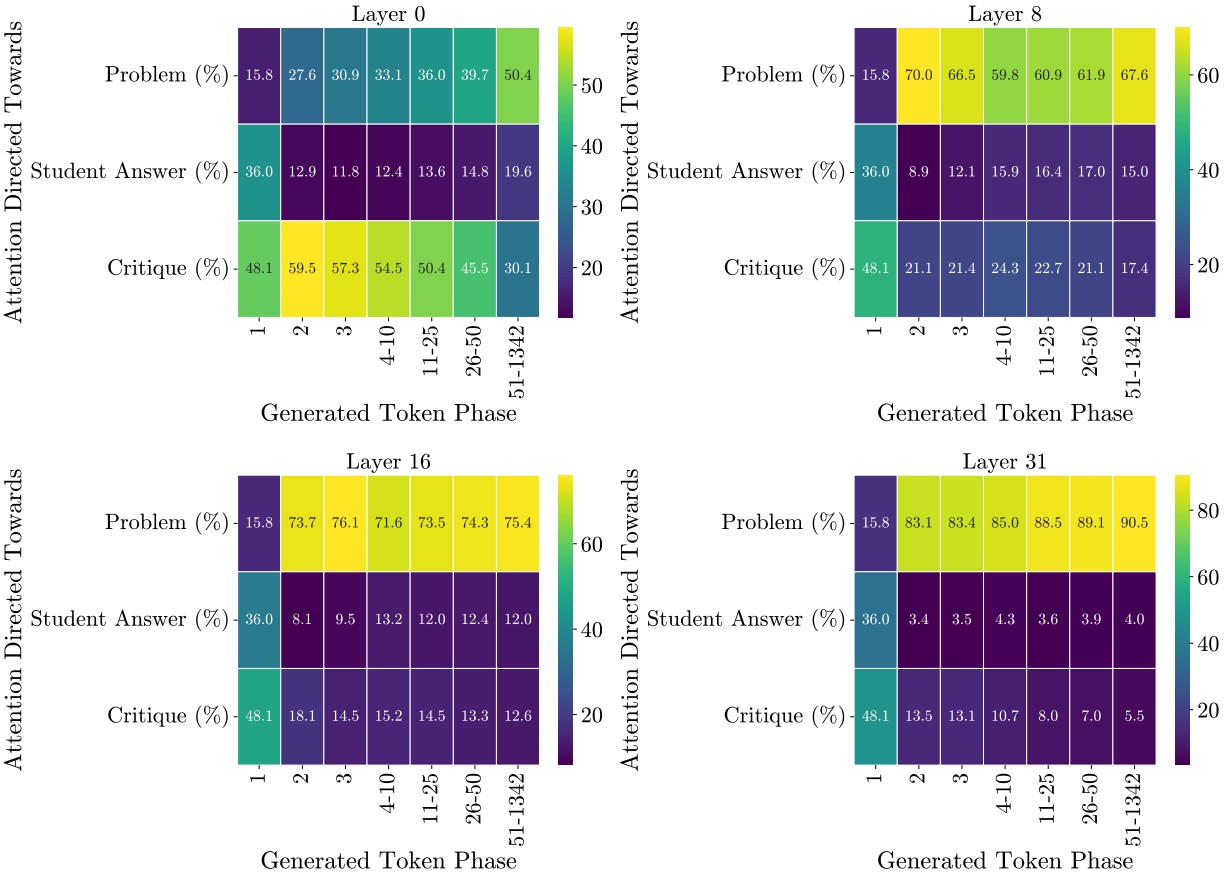

*Figure 10.* **Aggregated heatmap of attention by generation phase for representative layers.** The bright cells for the **Critique** in the first column for Layer 16 and 31 (48.1% and 45.5%) confirm that the initial planning phase is critique-driven, acting on the signal internalized by the early layers. The sustained brightness for the Critique in the Layer 0 heatmap illustrates its role in early-stage processing.

These results reinforce the conclusion that CGD improves the quality of individual solution attempts on extremely hard problems, and that its benefits generalize across competition years rather than being specific to a single benchmark instance.

## D.5 Robustness to Adversarial Critiques (Counterfactual Analysis)

To test whether CGD learns a functional skill of robust reasoning beyond simple contextual understanding, we performed a qualitative case study. We presented both the baseline Llama 3.1 Instruct model and our final CGD-trained model with a problem from our test set under two conditions. In the **Factual** condition, we provided the original, correct critique from our dataset. In the **Counterfactual** condition, we provided a generic but nonsensical critique that was irrelevant to the problem (e.g., incorrectly referencing a "quadratic formula" for a number theory problem). We then evaluated the correctness of the final generated answer in all four scenarios. Table 18 summarizes the results for a representative sample.

The results demonstrate a clear behavioral divergence. As shown in Table 18, the CGD model demonstrates **discriminative independence**. While the baseline SFT model is easily confused by the noise, the CGD model ignores the adversarial critique and generates the correct solution based on the problem statement alone. The baseline model proves brittle; it is functionally *derailed* by the nonsensical critique, failing to produce a correct answer. The CGD model, in contrast, exhibits superior robustness by **ignoring the misleading critique** and producing the correct answer based on its own problem analysis.

---

**Case Study: Counterfactual Analysis on Sample #42**

The goal of this analysis is to test for a functional understanding of critiques. The problem's gold-standard answer is **2310**.

---

**Baseline Model Analysis**
**Condition: Factual Critique**
**Correctness: CORRECT**
**Full Output:**

```
To find the smallest positive integer
↪   divisible by five distinct prime
↪   numbers, we need to understand the
↪   concept of prime numbers first...
...
2 * 3 * 5 * 7 * 11 = 2310
```

---

```
Therefore, the smallest positive integer
↪   divisible by five distinct prime
↪   numbers is 2310. This result is a
↪   direct consequence of the
↪   multiplication of the smallest prime
↪   numbers, which are the fundamental
↪   building blocks of all other
↪   numbers.
```

**Condition: Counterfactual Critique**
**Correctness: INCORRECT** (Derailed by the flawed critique.)
**Full Output:**

```
...A more accurate approach would be to
↪   understand that the smallest
↪   positive integer... is actually
↪   their least common multiple (LCM)...
To find the LCM of 2, 3, 5, 7, and 11,
↪   we can list the multiples of each
↪   number and find the smallest number
↪   that appears in all the lists:

Multiples of 2: 2, 4, 6, 10, 14, 22, 30,
↪   42, 70, 110, 154, 330,...
Multiples of 3: 3, 6, 9, 12, 15, 18, 21,
↪   24, 27, 30, 33, 36, 39, 42, 45, 48,
↪   51,..., 1542, 1545, 1548, 1551,
↪   1554, 1557, 1560, 1563, 1566, 1569,
```

---

**CGD-trained Model Analysis**
**Condition: Factual Critique**
**Correctness: CORRECT**
**Parsed Answer:** 2310
**Full Output:**

```
## Finding the Smallest Positive Integer
↪   Divisible by Five Distinct Prime
↪   Numbers

To find the smallest positive integer
↪   divisible by five distinct prime
↪   numbers, we need to follow a
↪   straightforward approach that
↪   involves identifying the five
↪   smallest distinct prime numbers and
↪   then multiplying them together. This
↪   method is based on the fundamental
↪   principle that prime numbers are the
↪   building blocks of all other
↪   numbers, and ...

### Definition of Prime Numbers

Before proceeding, it's essential to
↪   understand what prime numbers are.
↪   Prime numbers are natural numbers
↪   greater ...

### Identifying the Five Smallest
↪   Distinct Prime Numbers
```

*Table 18.* **Adversarial Robustness Test.** We presented models with a "Counterfactual Critique" containing false or irrelevant guidance. The Baseline model was derailed, incorporating the error. The CGD model successfully identified the critique as invalid, ignored it, and solved the problem correctly.

| Model | Outcome (Factual Critique) | Outcome (Counterfactual Critique) |
|---|---|---|
| LLaMA3.1-8B Instruct | CORRECT | INCORRECT (Derailed) |
| CGD | CORRECT | CORRECT (Robust) |

```
The first five prime numbers in
↪  ascending order are 2, 3, 5, 7, and
↪  11. These numbers are distinct,
↪  meaning they are all different from
↪  one another, ...

### Multiplication of the Prime Numbers

To find the smallest positive integer
↪  divisible by these five distinct
↪  prime numbers, we multiply them
↪  together:

2 * 3 * 5 * 7 * 11 = 2310

This calculation is straightforward and
↪  demonstrates the direct application
↪  of the initial logic. By multiplying
↪  the smallest prime numbers, ...

### Conclusion

Therefore, the smallest positive integer
↪  divisible by five distinct prime
↪  numbers is indeed 2310. This result
↪  is a direct consequence of
↪  multiplying the five smallest
↪  distinct prime numbers together. The
↪  approach taken is logical and
↪  mathematically sound, ...

In summary, the process of finding the
↪  smallest positive integer divisible
↪  by five distinct prime numbers
↪  involves identifying the five
↪  smallest prime numbers, which are 2,
↪  3, 5, 7, and 11, and then
↪  multiplying them together. This
↪  method ensures that the resulting
↪  product is the smallest possible
↪  integer that is divisible by each of
↪  the five distinct primes, making
↪  2310 the correct answer to the
↪  problem.
```

**Condition: Counterfactual Critique**
**Correctness: CORRECT**
**Parsed Answer:** 2310
**Full Output:**

```
To find the smallest positive integer
↪  divisible by five distinct prime
↪  numbers, we need to follow a
↪  systematic approach that involves
↪  ... However, the initial step
↪  involves understanding what prime
↪  numbers are and how to identify
↪  them.

Prime numbers are natural numbers
↪  greater than 1 that have no positive
↪  ...

The five smallest distinct prime numbers
↪  are indeed 2, 3, 5, 7, and 11. These
↪  are the first five prime numbers in
↪  the sequence of prime numbers.

Now, to find the smallest positive
↪  integer divisible by these five
↪  distinct prime numbers, ...

The calculation is as follows:
2 * 3 * 5 * 7 * 11 = 2310

Therefore, the smallest positive integer
↪  divisible by five distinct prime
↪  numbers is indeed 2310. This number
↪  is the product of the five smallest
↪  distinct prime numbers, ensuring
↪  that it is divisible by each of them
↪  without leaving a remainder.

In conclusion, the approach to finding
↪  the smallest positive integer
↪  divisible by five distinct prime
↪  numbers involves identifying the
↪  prime numbers, understanding their
↪  properties, and then calculating
↪  their product. The result, 2310, is
↪  the smallest number that meets the
↪  criteria of being divisible by five
↪  distinct prime numbers.
```

**Conclusion:** The CGD model exhibits superior robustness and instruction-following. It correctly solves the problem under both conditions while adhering to the specified output format. The baseline model, while capable of reasoning correctly, is brittle to both complex instructions and irrelevant, noisy feedback.

## D.6 Bayesian Interpretation

Finally, we interpret critique conditioning as a Bayesian posterior update. Let the student's initial output $y'$ define a prior distribution $p(y|x)$, and let the critique $c$ provide new evidence about correctness. The teacher's refinement can be viewed as a posterior distribution:

$$\underbrace{S_\theta(\hat{y} \mid x, y', c)}_{\text{Student posterior}} \propto \underbrace{T_\phi(c \mid x, y', \hat{y})}_{\text{Teacher likelihood}} \times \underbrace{S_{\text{init}}(\hat{y} \mid x, y')}_{\text{Student prior}}. \tag{3}$$

Here, $S_{\text{init}}(\hat{y} \mid x, y')$ is the student's original (prior) distribution over responses, while $T_\phi(c \mid x, y', \hat{y})$ acts as a scoring function that up-weights those $\hat{y}$ values better aligned with the critique. Note that the teacher "likelihood" need not be normalized; the proportionality sign indicates that normalization is implicit when forming the posterior.

In practice, CGD minimizes the KL divergence between the student's posterior and the teacher-defined target distribution, which directly implements Equation 3 in training via Algorithm 1. This interpretation highlights how critique guidance sharpens the student's prior into a more informative posterior, explaining the observed empirical gains.

## D.7 Evidence of Conceptual Internalization (Overlap Analysis)

To distinguish between *reasoning transfer* and *surface-level memorization*, we quantified the lexical dependence between the teacher's critiques and the student's refined answers. We analyzed 50,000 training examples from the WebInstruct dataset.

We computed two metrics:

- **Token-level overlap**: The percentage of unique tokens appearing in both the critique and the refined answer.

- **Bigram-level overlap**: The percentage of consecutive two-word sequences shared between the critique and the refined answer.

Our analysis reveals:

- **Token overlap (16.6%):** This moderate overlap reflects shared domain vocabulary (e.g., variables like $x, y$, terms like "integral", "mass"). This is expected, as the critique and solution must discuss the same subject matter.

- **Bigram overlap (5.7%):** This remarkably low phrase-level overlap confirms that the model is **not** copying the teacher's reasoning chains.

The high gap between token and bigram overlap indicates that CGD functions via **conceptual internalization**. The

student extracts the *semantic content* of the error (e.g., "forgot to divide by 2") from the critique but synthesizes the correction using its own generative process. This explains why the method generalizes effectively to inference time, where no critique is present.

# E Critique and Refinement Generation Prompts

For transparency and reproducibility, we provide the exact prompts used to generate critiques and refined answers from the teacher model during the CGD training data creation process.

## E.1 Critique Generation Prompt

> **Teacher Prompt for Generating Critiques**
>
> You are an expert in mathematics and reasoning. Your task is to carefully review a student's solution to a given problem and provide a detailed, constructive critique.
> **Problem:** {problem}
> **Student's Solution:** {student_answer}
> Please analyze the student's solution and provide a critique that:
>
> - Identifies any errors, misconceptions, or gaps in reasoning
>
> - Explains *why* these issues are problematic
>
> - Suggests the correct approach or concepts needed
>
> - Is clear, specific, and pedagogically helpful
>
> Your critique should focus on the reasoning process and help guide the student toward the correct solution without directly providing the full answer.

## E.2 Refined Answer Generation Prompt

> **Teacher Prompt for Generating Refined Answers**
>
> You are an expert in mathematics and reasoning. Given a problem, a student's initial attempt, and a critique of that attempt, provide a complete and correct solution that addresses all the issues identified in the critique.
> **Problem:** {problem}
> **Student's Initial Solution:** {student_answer}
> **Critique:** {critique}
> Please provide a refined, complete solution that:
>
> - Addresses all issues identified in the critique
>
> - Shows clear, step-by-step reasoning
>
> - Arrives at the correct answer
>
> - Maintains mathematical rigor and clarity

# F    Example CGD Training Data Sample

To better understand how **C**ritique-**G**uided **D**istillation enables improved response quality without format drift, we present a representative CGD training data sample. The CGD training pipeline involves three components: a student's initial answer, a teacher-generated critique that identifies specific flaws and explains the reasoning behind them, and a refined answer that incorporates the critique to correct the initial output. The full input tuple (prompt, student, critique, refined answer) is only used during training, and inference remains unchanged from standard decoding.

In the example, the student incorrectly assumes the number of electrons per nitrogen atom when calculating the mass of a nitride anion. The critique explicitly identifies this conceptual error, and the refined answer provides the correct computation with an explanation of the correct electronic configuration of the ion. This structured format teaches the model both the nature of the mistake and the reasoning needed to arrive at a correct solution, demonstrating how CGD effectively enhances model performance while preserving answer format consistency.

---

### CGD WebInstruct Training Data Sample

**Question:** Calculate the mass of a nitride anion that contains $2.408 \times 10^{24}$ nitrogen-based electrons.

**Student's Initial Answer:** To calculate the mass of the nitride anion, we first need to determine the number of nitrogen atoms in the anion. Since each nitrogen atom has 7 electrons, we can divide the total number of electrons by 7 to find the number of nitrogen atoms:

Number of nitrogen atoms $= \frac{\text{Total number of electrons}}{7} = \frac{2.408 \times 10^{24}}{7} = 3.443 \times 10^{23}$

The mass of a single nitrogen atom is approximately $14.007\,\text{u}$ (unified atomic mass units). We can multiply the number of nitrogen atoms by the mass of a single nitrogen atom to find the total mass of the nitride anion:

Mass of nitride anion $= 3.443 \times 10^{23} \times 14.007\,\text{u} = 4.835 \times 10^{25}\,\text{u}$

To convert the mass from unified atomic mass units to kilograms, we use the conversion factor $1\,\text{u} = 1.66053904 \times 10^{-27}\,\text{kg}$:

Mass in kg $= 4.835 \times 10^{25} \times 1.66053904 \times 10^{-27} = 8.033 \times 10^{-2}\,\text{kg}$

So, the mass of the nitride anion is approximately $8.033 \times 10^{-2}\,\text{kg}$ or $80.33\,\text{g}$.

**Critique:** The solution provided attempts to calculate the mass of a nitride anion based on the number of nitrogen-based electrons. Let's review the steps and calculations for accuracy and clarity.

1. **Determination of the Number of Nitrogen Atoms:** The solution starts by dividing the total number of electrons by 7 to find the number of nitrogen atoms. This assumes each nitrogen atom has 7 electrons, which is true for neutral nitrogen. However, the question refers to a *nitride anion*, typically $N^{3-}$, which has gained 3 electrons for a total of 10 electrons per atom. Thus, the correct calculation should be:

Number of nitrogen atoms $= \frac{2.408 \times 10^{24}}{10} = 2.408 \times 10^{23}$

2. **Calculation of the Mass:** Using the corrected number of nitrogen atoms and the same atomic mass:
Mass $= 2.408 \times 10^{23} \times 14.007\,\text{u} = 3.373 \times 10^{24}\,\text{u}$

3. **Conversion to Kilograms:**
Mass in kg $= 3.373 \times 10^{24} \times 1.66053904 \times 10^{-27} = 5.605 \times 10^{-3}\,\text{kg}$

The original solution incorrectly calculates the number of nitrogen atoms based on an incorrect assumption about the number of electrons per nitrogen atom in a nitride anion. This leads to an incorrect mass calculation. Conclusion: **Wrong.**

**Refined Answer: Step 1:** Determine the correct number of electrons per nitrogen atom in a nitride anion
A nitride anion ($N^{3-}$) has 10 electrons, which includes the 7 original electrons from the neutral nitrogen atom plus 3 additional electrons gained to form the anion.

**Step 2:** Calculate the number of nitrogen atoms in the nitride anion
To find the number of nitrogen atoms, divide the total number of electrons by the number of electrons per nitrogen atom in the anion, which is 10.

Number of nitrogen atoms $= \frac{\text{Total number of electrons}}{10} = \frac{2.408 \times 10^{24}}{10} = 2.408 \times 10^{23}$

**Step 3:** Calculate the mass of the nitride anion in unified atomic mass units (u)
The mass of a single nitrogen atom is approximately $14.007\,\text{u}$. Multiply the number of nitrogen atoms by the mass of a single nitrogen atom:

$$\text{Mass} = 2.408 \times 10^{23} \times 14.007 = 3.373 \times 10^{24}\,\text{u}$$

**Step 4:** Convert the mass of the nitride anion from unified atomic mass units (u) to kilograms
Use the conversion factor $1\,\text{u} = 1.66053904 \times 10^{-27}\,\text{kg}$:
Mass in kg $= 3.373 \times 10^{24} \times 1.66053904 \times 10^{-27} = 5.605 \times 10^{-3}\,\text{kg}$

The final answer is:

$$\boxed{5.605 \times 10^{-3}\,\text{kg}}$$

---

## F.1    Qualitative Comparison of Generated Responses

In the qualitative comparison of generated responses, the Critique Fine-Tuning (CFT) method exhibits a notable format drift, deviating from the expected answer format by producing *Conclusion* section and token at the end of each response. This behavior arises because CFT is trained to generate critiques, inadvertently leading the model to adopt a more elaborate response style. In contrast, the Critique-Guided Distillation (CGD) approach maintains the original response format while enhancing answer quality.

---

### Generated Answer after CFT

**Question:** True or false: there exists a graph with score (1, 1, 1, 2, 2, 3, 4, 5, 5). Answer with $\boxed{True}$ or $\boxed{False}$. (True or False)
**Answer:** Let's analyze the problem step by step.

---

Step 1: Understanding the problem
The problem asks whether there exists a graph whose score matches the given sequence (1, 1, 1, 2, 2, 3, 4, 5, 5). The score of a graph is determined by the number of vertices and edges in the graph. Specifically, the score is calculated as

score = number of vertices − number of edges.

Step 2: Analyzing the score sequence
The given score sequence is (1, 1, 1, 2, 2, 3, 4, 5, 5). This sequence represents the difference between the number of vertices and edges for a series of graphs. For example:

- A graph with score 1 could have 2 vertices and 1 edge.
- A graph with score 2 could have 3 vertices and 1 edge.

Step 3: Checking the feasibility of the sequence
To determine whether the sequence is valid, we need to verify if it is possible to construct a sequence of graphs where the score increases monotonically and matches the given sequence. This involves checking whether the number of vertices and edges can be adjusted to satisfy the score for each step.
Observations:

1. The sequence starts with 1, which is plausible for a graph with 2 vertices and 1 edge.

2. The sequence increases by 1 for the next three steps (1, 2, 3), which is consistent with adding edges to a graph while keeping the number of vertices constant.

3. The sequence then increases by 1 again (3, 4, 5), which is also consistent with adding edges to a graph while keeping the number of vertices constant.

4. The sequence ends with 5, which is plausible for a graph with 6 vertices and 1 edge.

**Key Check:** The sequence does not require any negative scores (i.e., more edges than vertices), which is a valid mathematical possibility. Therefore, the sequence is feasible.
Step 4: Conclusion
The given score sequence (1, 1, 1, 2, 2, 3, 4, 5, 5) is valid and corresponds to a sequence of graphs with increasing scores.
**Conclusion: right**

Step 2: Linearity The transformation $T(M) = \begin{bmatrix} 1 & 2 \\ 3 & 4 \end{bmatrix} M$ is clearly linear because matrix multiplication is distributive and associative. Thus, the linearity condition is satisfied.
Step 3: Injectivity To check injectivity, we need to verify whether $T(M_1) = T(M_2) \Rightarrow M_1 = M_2$. Let $M_1, M_2 \in \mathbb{R}^{2 \times 2}$. If $T(M_1) = T(M_2)$, then:

$$\begin{bmatrix} 1 & 2 \\ 3 & 4 \end{bmatrix} M_1 = \begin{bmatrix} 1 & 2 \\ 3 & 4 \end{bmatrix} M_2$$

This implies:

$$M_1 = M_2,$$

since the matrix $\begin{bmatrix} 1 & 2 \\ 3 & 4 \end{bmatrix}$ is invertible (its determinant is nonzero). Therefore, $T$ is injective.
Step 4: Surjectivity To check surjectivity, we need to verify whether for every $M \in \mathbb{R}^{2 \times 2}$, there exists $M' \in \mathbb{R}^{2 \times 2}$ such that $T(M') = M$. Let $M \in \mathbb{R}^{2 \times 2}$. We solve for $M'$ such that:

$$\begin{bmatrix} 1 & 2 \\ 3 & 4 \end{bmatrix} M' = M$$

This is equivalent to solving the system of linear equations:

$$\begin{bmatrix} 1 & 2 \\ 3 & 4 \end{bmatrix} \begin{bmatrix} x_1 \\ x_2 \end{bmatrix} = \begin{bmatrix} a & b \\ c & d \end{bmatrix},$$

where $M = \begin{bmatrix} a & b \\ c & d \end{bmatrix}$. The matrix $\begin{bmatrix} 1 & 2 \\ 3 & 4 \end{bmatrix}$ is invertible (its determinant is nonzero), so the system has a unique solution for any $M$. Therefore, $T$ is surjective.
Step 5: Conclusion Since $T$ is both injective and surjective, and it is linear, it is an isomorphism.
**Conclusion: right**

## Generated Answer after CFT

**Question:** Is the transformation $T(M) = \begin{bmatrix} 1 & 2 \\ 3 & 4 \end{bmatrix} M$ from $\mathbb{R}^{2 \times 2}$ to $\mathbb{R}^{2 \times 2}$ an isomorphism? Answer with $\boxed{True}$ or $\boxed{False}$. (True or False)
**Answer:** Step 1: Definition of an Isomorphism An isomorphism is a bijective linear transformation between two vector spaces. For $T(M) = \begin{bmatrix} 1 & 2 \\ 3 & 4 \end{bmatrix} M$ to be an isomorphism, it must satisfy the following conditions:
1. **Injectivity**: $T(M_1) = T(M_2) \Rightarrow M_1 = M_2$ 2. **Surjectivity**: For every $M \in \mathbb{R}^{2 \times 2}$, there exists $M' \in \mathbb{R}^{2 \times 2}$ such that $T(M') = M$ 3. **Linearity**: $T(aM_1 + bM_2) = aT(M_1) + bT(M_2)$ for all $a, b \in \mathbb{R}$ and $M_1, M_2 \in \mathbb{R}^{2 \times 2}$

## Generated Answer after CGD

**Question:** True or false: there exists a graph with score (1, 1, 1, 2, 2, 3, 4, 5, 5). Answer with $\boxed{True}$ or $\boxed{False}$. (True or False)

**Answer:** To determine if there exists a graph with the given score, we need to understand what a score in graph theory represents.
A score of a graph is the sequence of the degrees of its vertices, listed in non-increasing order. The given score is (1, 1, 1, 2, 2, 3, 4, 5, 5).

For a graph to exist with this score, it must be possible to construct a graph where the degrees of its vertices match this sequence. The sum of the degrees of all vertices in a graph is twice the number of edges (since each edge contributes to the degree count of two vertices). This sum is calculated as follows:

$$1 + 1 + 1 + 2 + 2 + 3 + 4 + 5 + 5 = 24$$

Since the sum of the degrees is 24, the number of edges in the graph must be 12 (because $24/2 = 12$).

Now, let's consider the degrees and how they can be distributed among vertices. The degrees are 1, 1, 1, 2, 2, 3, 4, 5, 5. To form

a graph, we need to ensure that the degrees can be assigned to vertices in such a way that the graph remains connected and that the degree of each vertex matches the given sequence.

Given the degrees, we can start by assigning the lowest degrees first. The three vertices with degree 1 must be connected to each other, as they cannot be isolated. This forms a triangle, which is a connected subgraph.

Next, we assign the degrees 2. These vertices can be connected to any of the existing vertices, including each other, as long as they are not isolated. Since we have two vertices with degree 2, they can be connected to the vertices with degree 1, or to each other, or to any of the higher-degree vertices.

The vertex with degree 3 can be connected to any of the existing vertices, including the ones with degree 2, or to the higher-degree vertices.

The vertices with degrees 4 and 5 can be connected to any of the existing vertices, including the ones with lower degrees, as long as they are not isolated.

Given these constraints, it is possible to construct a graph that matches the given score. For example, consider a graph with 9 vertices where:

1. Three vertices are connected in a triangle (degrees 1).

2. Two vertices are connected to each other and to one of the vertices in the triangle (degrees 2).

3. One vertex is connected to one of the vertices in the triangle and to one of the vertices with degree 2 (degree 3).

4. One vertex is connected to the vertices with degrees 1, 2, and 3 (degree 4).

5. Two vertices are connected to all other vertices, including the ones with degrees 1, 2, 3, and 4 (degrees 5).

This construction ensures that the degrees of the vertices match the given score, and the graph remains connected.

Therefore, the statement that there exists a graph with the score $(1, 1, 1, 2, 2, 3, 4, 5, 5)$ is true.

The final answer is: $\boxed{True}$

---

### Generated Answer after CGD

**Question:** Is the transformation $T(M) = \begin{bmatrix} 1 & 2 \\ 3 & 4 \end{bmatrix} M$ from $\mathbb{R}^{2 \times 2}$ to $\mathbb{R}^{2 \times 2}$ an isomorphism? Answer with $\boxed{True}$ or $\boxed{False}$. (True or False)

**Answer:** To determine if the transformation $T(M) = \begin{bmatrix} 1 & 2 \\ 3 & 4 \end{bmatrix} M$ from $\mathbb{R}^{2 \times 2}$ to $\mathbb{R}^{2 \times 2}$ is an isomorphism, we need to check if it satisfies the two properties of an isomorphism:
1. **Injectivity**: The transformation must be one-to-one, meaning that if $T(M_1) = T(M_2)$, then $M_1 = M_2$. 2. **Surjectivity**: The transformation must be onto, meaning that for every matrix $N \in \mathbb{R}^{2 \times 2}$, there exists a matrix $M \in \mathbb{R}^{2 \times 2}$ such that $T(M) = N$.

---

Let's examine the transformation $T(M) = \begin{bmatrix} 1 & 2 \\ 3 & 4 \end{bmatrix} M$.

Given $M = \begin{bmatrix} a & b \\ c & d \end{bmatrix}$, we have:

$$T(M) = \begin{bmatrix} 1 & 2 \\ 3 & 4 \end{bmatrix} \begin{bmatrix} a & b \\ c & d \end{bmatrix} = \begin{bmatrix} 1a + 2c & 1b + 2d \\ 3a + 4c & 3b + 4d \end{bmatrix}$$

Now, let's check for injectivity:
Suppose $T(M_1) = T(M_2)$. Then:

$$\begin{bmatrix} 1a_1 + 2c_1 & 1b_1 + 2d_1 \\ 3a_1 + 4c_1 & 3b_1 + 4d_1 \end{bmatrix} = \begin{bmatrix} 1a_2 + 2c_2 & 1b_2 + 2d_2 \\ 3a_2 + 4c_2 & 3b_2 + 4d_2 \end{bmatrix}$$

This implies that the corresponding elements of the two matrices are equal:

$$1a_1 + 2c_1 = 1a_2 + 2c_2$$
$$1b_1 + 2d_1 = 1b_2 + 2d_2$$
$$3a_1 + 4c_1 = 3a_2 + 4c_2$$
$$3b_1 + 4d_1 = 3b_2 + 4d_2$$

Simplifying, we get:

$$a_1 = a_2, \quad c_1 = c_2, \quad b_1 = b_2, \quad d_1 = d_2$$

Therefore, $M_1 = M_2$, and the transformation is injective.
Next, let's check for surjectivity:

Given any matrix $N = \begin{bmatrix} e & f \\ g & h \end{bmatrix} \in \mathbb{R}^{2 \times 2}$, we need to find a matrix $M = \begin{bmatrix} a & b \\ c & d \end{bmatrix}$ such that $T(M) = N$.
We have:

$$T(M) = \begin{bmatrix} 1a + 2c & 1b + 2d \\ 3a + 4c & 3b + 4d \end{bmatrix} = \begin{bmatrix} e & f \\ g & h \end{bmatrix}$$

This gives us the following system of equations:

$$1a + 2c = e$$
$$1b + 2d = f$$
$$3a + 4c = g$$
$$3b + 4d = h$$

Solving this system, we can find values for $a, b, c, d$ that satisfy the equations. For example, one possible solution is:

$$a = \frac{e}{1} = e, \quad b = \frac{f}{1} = f, \quad c = \frac{g - 3e}{4}, \quad d = \frac{h - 3f}{4}$$

Therefore, we can find a matrix $M$ that maps to any given matrix $N$, and the transformation is surjective.

Since the transformation $T(M) = \begin{bmatrix} 1 & 2 \\ 3 & 4 \end{bmatrix} M$ is both injective and surjective, it is an isomorphism.
The final answer is: $\boxed{True}$

# G  Quantitative Criterion for Critique Receptivity

Our experiments reveal that CGD's gains vary across model families (Appendix B.1.2). To allow practitioners to predict a base model's receptivity to CGD before committing to a full training run, we propose the **Relative Gradient Norm Reduction (RGNR)** heuristic.

**Definition.** Given a validation set $\mathcal{D}_{\text{val}}$ of $(x, y', c, \hat{y})$ tuples, we compute the gradient of the cross-entropy loss on the refined answer $\hat{y}$ under two conditions:

- **Baseline:** $g_{\text{base}} = \nabla_\theta \mathcal{L}(\hat{y} \mid x, y')$    (prompt + student answer only)

- **CGD:** $g_{\text{cgd}} = \nabla_\theta \mathcal{L}(\hat{y} \mid x, y', c)$    (full CGD context including critique)

The RGNR is defined as:

$$\text{RGNR} = \frac{\|g_{\text{base}}\|_2 - \|g_{\text{cgd}}\|_2}{\|g_{\text{base}}\|_2} \tag{4}$$

A positive RGNR indicates that the critique reduces the gradient norm, meaning the model's internal representations can efficiently route critique guidance toward producing better outputs even before any fine-tuning occurs.

**Empirical Measurements.** We measured RGNR on the base (pre-fine-tuning) versions of three student models used in our study:

*Table 19.* **RGNR measurements on base models prior to fine-tuning.** Higher RGNR correlates with larger downstream gains from CGD.

| Base Model | RGNR (%) | Downstream Math Gain |
|---|---|---|
| Mixtral-8x7B Instruct | 31.0 | +5.2% |
| LLaMA3.1-8B Instruct | 14.6 | +5.6% |
| OLMo-2-7B-Instruct | 10.8 | +3.4% |

**Interpretation Guidelines.**

- **Significant receptivity (RGNR $\geq$ 10%):** The base model's representations efficiently route critique guidance. All three models above meet this threshold and exhibit positive downstream gains.

- **Low receptivity (RGNR $<$ 5%):** A negligible reduction suggests the pre-training alignment does not leverage structured multi-turn feedback, serving as an early warning that the base model may not benefit substantially from CGD.

**Caveats.** RGNR is a practical heuristic, not a formal theoretical predictor. Our measurements span only three model families; future work should validate the correlation on a broader set of architectures and scales. Nevertheless, RGNR provides a computationally inexpensive diagnostic (requiring only a single forward and backward pass on a small validation set) that can inform resource allocation decisions before launching full CGD training.

# H  Critique Quality Analysis

To characterize the quality of critiques in our training data along dimensions beyond correctness, we conducted an LLM-as-judge evaluation of 20,000 training critiques.

**Setup.** We sampled 10,000 critiques for incorrect student answers and 10,000 critiques for correct student answers from our WebInstruct training data. Each critique was scored by `gpt-oss-120B` on two dimensions:

- **Informativeness** (1–5): Does the critique identify the specific error and suggest a path to correction? (5 = identifies exact error with fix suggestion; 1 = no useful information.)

- **Specificity** (1–5): Does the critique reference specific steps, quantities, or reasoning elements? (5 = references exact steps/quantities; 1 = entirely generic.)

**Main Results.** Table 20 summarizes the results. Critiques of incorrect student answers are significantly more informative and specific than those for correct answers, with large effect sizes.

*Table 20.* **LLM-as-judge evaluation of 20,000 training critiques.** Critiques of incorrect student answers are significantly more informative and specific.

| Metric | Wrong ($n$=10K) | Right ($n$=10K) | $\Delta$ | Cohen's $d$ |
|---|---|---|---|---|
| Informativeness | 4.36 (0.62) | 3.71 (0.63) | +0.65 | 1.04 (large) |
| Specificity | 4.51 (0.55) | 4.16 (0.55) | +0.36 | 0.65 (medium) |
| High quality (both $\geq$ 4) | 93.7% | 70.8% | +22.9pp | OR=6.10 |

The informativeness distribution reveals an interesting asymmetry: 42.3% of wrong-answer critiques achieve the maximum informativeness score of 5, compared to only 4.2% for right-answer critiques, which is a $10.1\times$ ratio. This confirms that the teacher naturally produces substantially richer feedback when the student makes genuine errors.

**Length Confound Control.** To rule out the possibility that the informativeness gap is driven by critique verbosity, we stratified critiques into five length quintiles and computed the gap within each:

The mean within-quintile gap (+0.65) matches the overall gap, confirming that critique length does not explain the informativeness difference.

*Table 21.* **Informativeness gap by critique length quintile.** The gap holds across all quintiles, ruling out verbosity as a confound.

| Length Quintile | Wrong Inf. | Right Inf. | Gap |
|---|---|---|---|
| Q1 (shortest) | 4.36 | 3.72 | +0.64 |
| Q2 | 4.41 | 3.70 | +0.71 |
| Q3 | 4.38 | 3.69 | +0.70 |
| Q4 | 4.36 | 3.72 | +0.64 |
| Q5 (longest) | 4.31 | 3.72 | +0.59 |

**Judge Calibration.** To validate that the judge discriminates meaningfully, we scored 600 control critiques (200 per condition):

*Table 22.* **Judge calibration.** The judge assigns high scores to real critiques and appropriately low scores to controls.

| Condition | Informativeness | Specificity |
|---|---|---|
| Real critiques | 4.26 | 4.39 |
| Generic platitudes ("Try again") | 1.74 | 1.04 |
| Unrelated problem critiques | 1.00 | 1.18 |

The >2.5-point separation between real critiques and controls confirms that the judge is not rubber-stamping and can meaningfully distinguish informative feedback from noise.

## I  Critique Content Ablation

This section provides additional details for the critique content ablation presented in Table 4 of the main text.

**Setup.** From a subset of ∼10K problems, we created three training sets with **identical** student answers and reference outputs, varying **only** the critique text:

- **Condition A (Specific & Relevant):** Original high-quality critiques that identify the exact error in the student's answer.

- **Condition B (Specific but Irrelevant):** Real critiques drawn from *unrelated* problems. These are well-formed and specific, but address errors the student did not make.

- **Condition C (Generic):** A fixed string: "The answer is incorrect. Please review and provide a corrected solution."

All other variables (problems, student answers, refined answer targets, hyperparameters) are held constant. Results are averaged over 3 random seeds.

**Extended Results.** In addition to the benchmarks reported in the main text, we also evaluated on GSM8K: A (85.5±0.2), B (85.6±0.1), C (85.7±0.1). All three conditions perform near the base model's ceiling (85.3%), leaving no room for differentiation on this easier benchmark.

**Key Findings.**

- **Specificity matters (A vs. C):** Specific relevant critiques outperform the generic baseline on reasoning-intensive benchmarks: +1.5 on MATH-500, +5.0 on AMC23, and +4.4 on AIME24. GSM8K shows no separation, as all conditions already perform near the base model's ceiling.

- **Relevance matters (B vs. C):** Irrelevant critiques perform *worse* than the generic baseline on AMC23 (18.3 vs. 26.7) and AIME24 (6.7 vs. 7.8). When critique content systematically contradicts the problem, it degrades learning beyond what a content-free placeholder causes.

- **Active conditioning:** The ordering A > C > B on hard benchmarks demonstrates that the model reads and internalizes critique content during training, rather than treating it as an inert padding token.

This training-time sensitivity is complementary to the inference-time robustness reported in Appendix D.5: a model *already trained* on high-quality critiques can discard adversarial noise at test time (Table 18), but training on misleading critiques prevents effective learning in the first place.

Combined with the performance drop when critiques are removed entirely (Figure 3), the adversarial noise ablation (Table 18), and the attention flow analysis (Appendix D), these results establish a clear quality–performance relationship across five conditions: no critique, noise, generic, irrelevant, and specific.

**Note on Scale.** This ablation uses a ∼10K training set (smaller than the 100K used in our main experiments) for computational efficiency. The pattern is expected to hold at full scale, as the critique content mechanism does not depend on dataset size.

## J  Code Instructions

The full codebase, including data generation, training, and evaluation scripts, is publicly available at https://github.com/CapitalOne-Research/Critique-Guided-Distillation. This package provides all necessary components to reproduce our key results and facilitate further experimentation. Specifically, it includes:

- **Critique Generation and Refinement:** Scripts for generating critiques from model outputs and refining answers using these critiques.

- **Evaluation Codebase:** End-to-end evaluation pipelines for computing exact match accuracy and other relevant metrics across multiple benchmarks.

- **LLaMA-Factory Integration:** The LLaMA-Factory framework to support Supervised Fine-Tuning (SFT).

- **Configuration Files and Run Scripts:** YAML/JSON config files and shell scripts used to launch experiments across various model architectures and tasks.

Instructions for installing dependencies, setting up the environment, and running end-to-end training and evaluation pipelines are included in the README.md file within the archive.

