# OpenReview forum: "Critique-Guided Distillation for Robust Reasoning via Refinement"
_ICML.cc/2026/Conference — ICML 2026 regular_

### Official Review · Reviewer_W58V · 2026-02-24

**Soundness:** 3
**Presentation:** 3
**Significance:** 3
**Originality:** 3
**Overall Recommendation:** 4
**Confidence:** 3

**Summary:**

The paper introduces CGD a novel fine-tuning framework aimed at improving the reasoning capabilities of language models. Unlike traditional methods like Critique Fine-Tuning, which train models to generate critiques, CGD decouples the process by training the model to use teacher-generated critiques to refine its own responses. The critiques are used during training to guide the model's learning but are not present during inference, thus ensuring no additional computational overhead at inference time.

**Compliance With Llm Reviewing Policy:**

Affirmed.

**Final Justification:**

This paper introduces a novel critique-guided distillation framework that improves reasoning capabilities of language models without adding inference overhead. The method effectively avoids issues present in standard Critique Fine-Tuning, including output-format drift and catastrophic forgetting, and demonstrates robust gains across multiple benchmarks, including low Pass@k settings and out-of-distribution tasks like code generation. The rebuttal convincingly addressed concerns regarding performance when student and teacher models are similar, robustness to noisy critiques, and computational efficiency compared to RL-based approaches, providing both mechanistic insights and empirical evidence. Overall, CGD presents a sound, original, and practically significant contribution, and the rebuttal reinforces my confidence in its value, leading me to maintain a positive recommendation

**Key Questions For Authors:**

1.How does CGD perform when the student and teacher models have similar capabilities? In cases where the teacher model is close to the optimal solution, what are the expected performance gains for the student model during training?

2.How do you ensure the quality of critiques in cases where the teacher model is not perfect, or when the critiques are noisy? Could CGD still function effectively if critiques are not always accurate, and how does the method handle this scenario?

3.While CGD is computationally efficient at inference time, it requires additional data generation during training for critiques and refined answers. How do you assess the trade-off between the benefits of CGD and the computational overhead？

**Limitations:**

yes

**Strengths And Weaknesses:**

**Strengths**

1.This paper introduces a novel paradigm by training models to use critiques for refining their responses. It avoids issues like output-format drift and catastrophic forgetting that arise in Critique Fine-Tuning.

2.CGD effectively enhances the reasoning abilities of models across multiple benchmark tasks, including mathematical reasoning and general problem-solving tasks. It demonstrates improvements in reasoning quality per sample rather than relying on extensive sampling, especially in low Pass@k conditions.

3.CGD avoids the degradation in general instruction-following abilities seen in CFT. It maintains strong performance on instruction-following benchmarks and transfers effectively to out-of-distribution tasks, such as code generation, despite being trained on non-code data.

**Weaknesses**

1.CGD heavily relies on the quality of critiques. If the teacher's critiques are not accurate or detailed enough, the student model may not learn effectively, limiting the potential improvements in reasoning.

2.CGD introduces some data generation overhead for teacher critiques and refined answers, which could become nontrivial at scale.

3.The paper lacks a comprehensive comparison with RL approaches such as GRPO.

---

> ### Author Rebuttal · Authors · 2026-03-27
>
> We thank the reviewer for recognizing how CGD avoids the format drift and capability degradation inherent in CFT.
>
> **1. Performance When Student and Teacher are Similar**
> "Similar capabilities" are explored in two ways in our results:
>
> * **Narrowing the capability gap:** In Table 3, we distilled Qwen2.5-Math-7B using S1.1-32B (vastly smaller than GPT-4o/Claude). Despite this narrower gap, CGD achieved a 49.0% avg (**+21.2% absolute gain** over base), outperforming the GPT-4o-taught CFT baseline. This proves gains stem fundamentally from the critique-refinement structure, not just raw knowledge transfer.
> * **Task-specific capabilities:**
>   * *Scenario A (Both models struggle):* On **GPQA** (Table 9), the base student scores 30.8% and the larger teacher (LLaMA3.3-70B) struggles at 37.9%. CGD pushes the student to 35.9%, capturing almost the entire gap. Providing a struggling teacher with a targeted prompt forces a focused verification role, extracting a high-value signal.
>   * *Scenario B (Both models are near optimal):* On **GSM8K** (Table 1), the LLaMA3.1-8B student is already near-optimal (85.3%). When capabilities are similar and the student is near optimal, forcing the model to generate critiques (as in the CFT baseline) actually *degrades* performance to 84.8%. In contrast, CGD acts as a safe scaffold, preserving and slightly improving the score to 85.7%.
>
> Thus, when capabilities are similar, gains are naturally smaller, but **CGD safely distills verification logic and avoids the capability degradation observed in standard critique generation**.
>
> **2. Handling Noisy or Imperfect Critiques**
> CGD treats imperfect critiques as informative guidance rather than strict logical constraints overriding mathematical reality. Robustness comes from two mechanisms:
>
> * **The Loss Objective (Eq. 1):** Computed solely on the correct Refined Answer. This discourages blindly propagating flawed reasoning from noisy critiques, as the final output must align with the mathematically correct target.
> * **Data Mixture (Fig 4):** We explicitly train the model on a 50/50 mixture of correct and incorrect student answers. This balanced exposure forces the model to develop an internal discriminator, encouraging it to actively evaluate the critique's validity rather than assuming every critique requires a correction.
>
> We verified this robustness both mechanistically and empirically:
> * **Mechanistically (Appendix D.3):** Early "planning" layers attend heavily to the critique (48.1% at Layer 0), but final "execution" layers (Layer 31) pivot to >90% focus on the Problem statement, allowing the model to mathematically discard flawed guidance.
> * **Empirically (Appendix D.5):** We gave a nonsensical adversarial critique (Tab 16) to the CGD model, it demonstrated "discriminative independence", ignoring the noise and generating the correct answer based on the prompt alone.
>
> **3. Assessing the Computational Trade-off**
> Our total pipeline (~48 A100 GPU-hours, Tab 11) is exceptionally efficient compared to RL paradigms (e.g., 1,152 H100 hours). At deployment, multi-pass methods (Self-Refine) require 2-3 autoregressive passes per query, multiplying inference latency/FLOPs. At scale, CGD's single-pass inference savings far outweigh the initial generation investment.
>
> **4. Comparison with RL approaches (e.g., GRPO)**
> We appreciate the suggestion. In Appendix B.4, we explicitly compare CGD against SimpleRL-Zero (uses GRPO), a state-of-the-art RL-based reasoning framework. While RL methods like GRPO excel at exploring novel solution trajectories (higher AIME scores), they require immense compute (e.g., 1,152 H100 hours for SimpleRL vs. 48 A100 hours for CGD). We view CGD and GRPO as complementary: CGD is a highly efficient supervised distillation paradigm that can provide a mathematically robust, self-correcting initialization model, which can then be further optimized via GRPO. We are leaving this as an open question for future refinement.

---

> > ### Author Rebuttal · Reviewer_W58V · 2026-04-01
> >
> > Thank you for the detailed response and clarifications. My concerns have been addressed, and I will maintain my positive score of 4.

---

### Official Review · Reviewer_CefB · 2026-03-11

**Soundness:** 3
**Presentation:** 3
**Significance:** 3
**Originality:** 3
**Overall Recommendation:** 4
**Confidence:** 4

**Summary:**

This paper introduces Critique-Guided Distillation (CGD), a novel fine-tuning framework designed to enhance the complex reasoning capabilities of large language models. Unlike existing methods that train models to explicitly generate critiques during inference, CGD leverages teacher-generated critiques solely as an auxiliary conditioning signal during training to refine flawed student responses. By decoupling critique consumption from generation, the method internalizes error-correction behaviors without suffering from format drift or multi-turn inference overhead. The core contributions include demonstrating consistent performance gains on rigorous mathematical benchmarks across various model architectures while preserving general instruction-following capabilities.

**Compliance With Llm Reviewing Policy:**

Affirmed.

**Final Justification:**

After considering both the paper and the authors’ rebuttal, my overall assessment remains unchanged. I find the paper technically solid, clearly presented, and relevant to an important problem in reasoning-oriented language model training. In particular, the central idea of using critique signals only during training is a thoughtful and original design choice, and the empirical results suggest that this approach can improve reasoning performance while largely preserving general instruction-following ability. At the same time, I feel that there is still room for improvement in terms of the method’s cost efficiency and in how teacher and student models should be selected in practice, so the overall approach is not yet fully mature. Therefore, I maintain my current evaluation. I view the paper as reasonably original, although the core idea is not particularly paradigm-shifting. For these reasons, I believe the paper makes a worthwhile contribution and supports acceptance, though its overall impact is somewhat limited.

**Key Questions For Authors:**

1. The manuscript claims that CGD introduces no inference-time overhead compared to multi-turn reasoning methods. However, Table 4 shows a 4.4-fold increase in the number of generated words for the AIME 2024 benchmark. Since autoregressive decoding cost scales directly with sequence length, this increased token count raises actual latency and computational cost. Could the authors provide a direct wall-clock time or total FLOPs comparison between a single-pass CGD generation and a multi-turn baseline like Self-Refine?
2. In Appendix B.1.2, the authors provide an explanation for the performance gain variations across model families (e.g., the substantial leaps for LLaMA/Qwen versus the marginal gains for Mixtral/OLMo). Attributing this discrepancy to differences in "alignment data quality" and "pre-training inductive biases" is well-supported by existing literature. However, this reasonable explanation leaves the applicability of CGD on novel architectures somewhat difficult to predict in practice. To further enhance the practical utility and broad impact of this work, could the authors explore or suggest a quantitative criterion for practitioners to assess a base model's "receptivity" to critiques before applying CGD?
3. In Appendix G, the authors mention a code archive (code.zip) included in the supplementary materials for reproducibility. However, no supplementary files were found in the submission system. Could the authors provide an anonymous link to this codebase or submit the missing materials during the rebuttal phase to facilitate a complete assessment of the work's reproducibility?

**Limitations:**

yes

**Strengths And Weaknesses:**

1. Soundness
The submission is technically sound and supported by comprehensive experiments across multiple model families. The authors provide thorough internal mechanism analyses such as attention visualization and counterfactual testing to validate their claims. However, the authors state in the appendix that a code archive is provided for reproducibility, but this file was not included in the supplementary materials.
2. Presentation
The authors clearly position their work within the context of recent literature and explicitly differentiate their approach from earlier methods. They also effectively articulate complex theoretical intuitions using straightforward language.
3. Significance
This paper addresses a highly relevant problem by exploring how to distill deep reasoning capabilities without compromising general instruction-following skills. The proposed method demonstrates substantial improvements on benchmarks and provides a practical training paradigm.
4. Originality
The originality of this work lies in its clever decoupling of critique generation and critique consumption during the training phase. While utilizing external feedback is a known concept in machine learning, applying it to solve the format drift problem in model distillation is innovative. This creative combination of existing techniques offers valuable new perspectives on reasoning enhancement.

---

> ### Author Rebuttal · Authors · 2026-03-27
>
> Thank you for your strong support and for highlighting the originality of decoupling critique consumption from generation.
>
> **1. Inference Overhead vs. Autoregressive Latency**
> When we stated that CGD introduces "no additional overhead", we strictly meant *architectural* or *system-level* overhead. Unlike Self-Refine or Reflexion, CGD does not require multi-pass decoding loops, external reward models, or tree-search algorithms at inference time. However, you are completely correct that the 4.4x longer reasoning chains observed on AIME 2024 (Table 4) inherently increase autoregressive decoding latency. This internalized self-correction shows itself as a more deliberate, step-by-step internal search. Higher accuracy comes with the computational trade-off of generating more tokens per pass (at least for the complext AIME task we have looked at, note that it may not need to do so for some other less complex tasks). We will update the manuscript to explicitly state "no *architectural* inference-time overhead," while acknowledging the increased token generation length as a natural latency trade-off.
>
> **2. Quantitative Criterion for Critique Receptivity**
> This is a great suggestion. Based on our diagnostic analyses (Appendix D.1), we propose a formalized heuristic: **Relative Gradient Norm Reduction (RGNR)**. By comparing the gradient norm of the cross-entropy loss on the `Refined Answer` for the prompt-only baseline ($g_{base}$) against the critique-conditioned prompt ($g_{cgd}$), one can calculate how much the critique stabilizes the optimization path. We emphasize that RGNR is a heuristic and a practical guideline, rather than a formal theoretical predictor.
>
> **During this rebuttal phase**, we measured this heuristic on the base models used in our study prior to any fine-tuning:
> * **Mixtral-8x7B Instruct (Base):** RGNR of 31.0%, correlating with large absolute downstream gains (+5.2% avg on Math reasoning).
> * **LLaMA3.1-8B Instruct (Base):** RGNR of 14.6%, similarly correlating with large downstream gains (+5.6% avg on Math reasoning).
> * **OLMo-2-7B-Instruct (Base):** RGNR of 10.8%, correlating with its comparatively lower, though still positive, downstream gains (+3.4% avg on Math reasoning).
>
> While this is a heuristic, a significant drop (>10%) acts as a strong positive indicator of a model's receptivity, whereas a smaller drop (<5%) signals that a base model struggles to route multi-turn feedback. We will formalize this in a new Appendix H:
>
> **Appendix H: Quantitative Criterion for Critique Receptivity**
> To allow practitioners to predict a base model's receptivity to CGD before committing to a full training run, we formalize Relative Gradient Norm Reduction (RGNR).
> *Procedure:* Sample a validation set $\mathcal{D}_{val}$ of $(x, y', c, \hat{y})$.
>
> Compute gradients for the target $\hat{y}$ given only the prompt and initial answer ($g_{base}=\nabla_\theta\mathcal{L}(\hat{y}|x,y')$). Next, compute the gradient given the full CGD context ($g_{cgd}=\nabla_\theta\mathcal{L}(\hat{y}|x,y',c)$). Calculate: $RGNR=(\lVert g_{base}\rVert_2-\lVert g_{cgd}\rVert_2)/\lVert g_{base}\rVert_2$.
>
> *Interpretation:*
> * **Significant Receptivity ($RGNR \ge 10\%$):** Indicates the base model's internal representations efficiently route the critique's semantic guidance prior to any training. Mixtral (31.0%), LLaMA (14.6%), and OLMo (10.8%) demonstrated significant RGNR, correlating with observed downstream absolute gains (+5.2%, +5.6%, and +3.4% respectively).
> * **Low Receptivity ($RGNR < 5\%$):** A negligible drop suggests the pre-training alignment ignores structured multi-turn feedback, serving as an early warning that the base model may not benefit significantly from critique-guided distillation.
>
> *Note: These measurements were conducted during the rebuttal phase and will be included in the camera-ready version.*
>
> **3. Missing `code.zip` File**
> We sincerely apologize for the upload error. Due to an oversight during the final upload window, the file did not attach successfully. Under current ICML policies, we are strictly prohibited from providing external links to repositories. However, please be assured the full codebase is prepared and will be released upon acceptance. All training hyperparameter details and prompts required for reproduction are fully documented in Appendix A and E.

---

> > ### Author Rebuttal · Reviewer_CefB · 2026-04-02
> >
> > The authors have addressed my concerns. Accordingly, I maintain my positive recommendation.

---

### Official Review · Reviewer_ec1s · 2026-03-15

**Soundness:** 2
**Presentation:** 3
**Significance:** 2
**Originality:** 2
**Overall Recommendation:** 4
**Confidence:** 3

**Summary:**

This paper introduces Critique-Guided Distillation (CGD), a training framework that teaches LLMs to correct and refine their reasoning without suffering from format drift. During training, the student model learns to correct its own flawed initial answers conditioned on previous flawed reasoning and teacher's feedback. At inference time, it generates the response directly in a single pass. Empirically, CGD consistently outperforms SFT and CFT baselines across multiple model families and demonstrates improvements on several mathematical reasoning benchmarks.

**Compliance With Llm Reviewing Policy:**

Affirmed.

**Final Justification:**

The clarifications address my concerns on the fair comparison.

**Key Questions For Authors:**

See weaknesses.

**Limitations:**

yes

**Strengths And Weaknesses:**

Strengths:
- The empirical results show promising improvements over strong baselines including SFT, distilled SFT, and Critique Fine-Tuning (CFT) across several reasoning benchmarks and model families.
- The diagnostic analyses are interesting: by tracking gradient norms, and layer-wise attention flow, the authors provide convincing mechanistic evidence that the student model functionally internalizes the reasoning rather than just memorizing critique tokens.

Weaknesses:
- The training setup raises concerns regarding fairness: the authors state that *All experiments are trained on 16 Nvidia A100 GPUs for 30 minutes, amounting to 8 A100 GPU-hours per experiment, using identical data splits and hyperparameters across methods*. However, such a training configuration is somewhat unusual in practice and may introduce unintended biases. It is unclear how many training examples or tokens each baseline actually processes under this setting. For example, the critique may be longer than the refined answer, as a results, CGD may have more training steps than CFT, making the comparison unfair.
- CGD requires the teacher model to generate both a critique and a refined answer for each example. This introduces additional computational cost during data generation compared to methods such as CFT, which only require critique generation.
- The conceptual novelty of the approach appears somewhat limited relative to prior work on critique-based learning and refinement methods. A more explicit discussion of how CGD differs from and improves upon these existing methods would strengthen the paper.

---

> ### Author Rebuttal · Authors · 2026-03-27
>
> We appreciate your feedback and are glad you found the diagnostic analyses convincing. We address your concerns below and clarify a phrasing error in our training setup.
>
> **1. Fairness of the Training Setup (Epochs vs. Time)**
>
> We sincerely apologize for the confusing phrasing in Section 4.1.2 where we stated at line 252 "All experiments are trained on 16 Nvidia A100 GPUs for 30 minutes, amounting to a total of 8 A100 GPU-hours per experiment, using identical data splits and hyperparameters across methods (see Appendix A for more details)." You are completely right to raise a fairness concern.
>
> **We want to clarify that time was not a stopping condition; no baseline received more training steps.** All methods were trained under strictly identical conditions (Appendix A.2, Table 6, and Section 4.1.1):
> - **Dataset size:** 100K examples (identical splits)
> - **Training duration:** Exactly 1 epoch
> - **Global batch size:** 64
> Thus, every baseline processed exactly 100,000 examples over an identical number of optimization steps.
>
> **Why CGD's design is a fair comparison:**
> While CGD inputs include critiques (and thus process more tokens per example), this richer context is the core methodological contribution we aim to evaluate. Note that CFT similarly processes longer sequences than standard SFT (includes initial answer + critique); CGD extends this by conditioning on the refined answer, the trajectory of correction itself.
>
> Critically, controlled ablations prove gains arise from semantic critique content, not input length:
> - Removing the critique drops performance (Figure 3), despite reducing sequence length.
> - Replacing critiques with adversarial noise of similar length yields no benefit (Table 16).
>
> This proves gains arise from the critique's guidance, not from processing additional tokens.
>
> The "30 minutes" simply observed wall-clock efficiency. We will revise Section 4.1.2 to explicitly state the epoch-bounded protocol.
>
> **Visual and quantitative evidence of training fairness:**
> * **Figure 7 (Appendix B.3):** Both methods undergo exactly 1 epoch. CGD exhibits a smoother optimization trajectory, indicating the critique provides more consistent supervision, not additional training steps.
> * **Table 10 (Appendix B.3):** CGD exhibits superior stability across learning rates. At LR=5e-6, CFT drops 9.2 points (39.3 to 30.1), while CGD drops only 2.6 points (44.8 to 42.2).
>
> This robustness stems from CGD's training objective: by optimizing the refined answer rather than critique generation, CGD avoids the format drift that makes CFT non-stable at higher learning rates (see Appendix B.2.1 for detailed analysis).
>
> **2. Data Generation Cost Trade-offs**
>
> CGD requires generating both critique and refined answer, introducing upfront cost (acknowledged in Section 5). However, we view this as a favorable one-time investment. While CFT has marginally cheaper offline generation, training models to generate critiques causes a **catastrophic -21.3% format drift degradation on IFEval (Table 2)**. CGD invests slightly more compute offline, but preserves general capabilities, achieves SOTA single-pass reasoning, and completely eliminates the massive recurring computational overhead of multi-pass inference at deployment. As shown in Table 11, our total training and data generation compute (48 A100 hours) still remains drastically lower than RL-based reasoning paradigms (1,152 H100 hours) while providing similar performance improvements.
>
>
> **3. Conceptual Novelty Relative to Prior Work**
>
> We appreciate the opportunity to clarify our fundamental contribution. While critique-based learning is established, prior works suffer from distinct limitations:
> - **Standard Distillation (ORCA):** Ignores the student's specific failure modes.
> - **Outcome-based RL:** Provides scalar rewards lacking dense feedback explaining *why* a step is wrong.
> - **Multi-pass Refinement (Self-Refine):** Requires the model to generate its own critique at deployment, multiplying inference latency.
> - **Critique Fine-Tuning (CFT):** Forces the model to write critiques during training, causing severe format drift.
>
> **Our core conceptual novelty is fundamentally decoupling critique *consumption* from critique *generation*.** In existing refinement methods, a model must autoregressively generate a critique to benefit from it. CGD treats critique purely as a **training-time semantic scaffold**. By conditioning the refinement on the critique but calculating loss *only* on the final answer, CGD encourages the student to internalize the reasoning trajectory required to correct its own specific errors.
>
> This strictly improves upon prior methods by yielding the deep reasoning benefits of multi-turn refinement within a **single-pass inference model**, effectively bridging the gap between standard distillation and RL without breaking instruction-following. We will prominently add this explicit framing to the Introduction.

---

> > ### Author Rebuttal · Reviewer_ec1s · 2026-04-04
> >
> > Thank the authors for the clarifications, I have raised my score.

---

### Official Review · Reviewer_zXkr · 2026-03-17

**Soundness:** 3
**Presentation:** 2
**Significance:** 3
**Originality:** 3
**Overall Recommendation:** 5
**Confidence:** 3

**Summary:**

This paper introduces Critique-Guided Distillation (CGD), where during finetuning, the student is trained to refine flawed responses conditioned on teacher critiques. Critiques serve as training-time-only supervision signals and are absent at inference, requiring only a single forward pass with no additional overhead. The authors evaluate CGD across multiple model families and benchmarks, demonstrating improvements over Critique Fine-Tuning (CFT) and standard SFT distillation on mathematical reasoning tasks while preserving general instruction-following capabilities.

**Compliance With Llm Reviewing Policy:**

Affirmed.

**Ethical Review Concerns:**

No ethical concerns.

**Final Justification:**

The author's response resolved my concerns.

**Key Questions For Authors:**

- Regarding table 2 and table 3, can you provide results where CGD and CFT use identical teacher models to enable fair comparison?
- What is the sensitivity of CGD to critique quality (beyond correctness)? Have you experimented with critiques of varying specificity or length?
- Why were S1.1 models chosen over more standard Qwen2.5/Qwen3 models for the Qwen family experiments?

**Limitations:**

Yes, this paper has included a Limitations section.

**Strengths And Weaknesses:**

## Strengths
- The key insight of using critiques as training-time scaffolding rather than requiring the model to generate critiques is clever.
- Comprehensive ablation studies. The paper includes thoughtful ablations on critique correctness mixtures (Figure 4), learning rate sensitivity (Table 5), and the role of critique conditioning (Figure 3), providing insights into why CGD works.





## Weaknesses

1. **Inconsistent experimental setup undermines fair comparisons**:
   - In Table 2, CFT uses GPT-4o while CGD uses LLaMA3.3-70B Instruct as the teacher. The claim that "CFT severely degrades general capabilities" may not hold if the degradation stems from the different teacher model rather than the training paradigm itself.
   - In Table 3, CFT uses GPT-4o while CGD uses Claude Sonnet 3.7 or S1.1-32B. Without controlling for teacher model, we cannot attribute performance differences to the method rather than teacher quality.
   - **Recommendation**: Conduct controlled experiments where CGD and CFT use the same teacher model across all comparisons.

2. **Unconventional model family choices**: The Qwen family experiments use S1.1-3B/32B models, which are less commonly used in the community and are specifically math-specialized. This choice may inflate CGD's apparent gains since the training data heavily overlaps with the evaluation domain. Using Qwen2.5 models (as in the original CFT paper) or Qwen3 series would make more sense.

3. **Evaluation protocol unclear**: The main tables do not consistently specify whether results are greedy decoding or pass@k. Table 4 clarifies this distinction for AIME, but the evaluation protocol for other tables should be explicitly stated. Additionally, Table 4 does not specify which model is being evaluated.

4. **Critique quality dependency not fully explored**: The paper acknowledges that CGD "assumes access to informative critiques" but does not systematically study how critique quality affects downstream performance. The ablation on critique correctness mixture (Figure 4) is helpful but focuses on student answer correctness rather than critique informativeness or specificity.

---

> ### Author Rebuttal · Authors · 2026-03-29
>
> We thank the reviewer for their feedback and for acknowledging the strength of our ablation studies. We address your concerns below:
>
> **1. Identical Teacher Models for CGD and CFT (Tables 1, 2 & 3)**
> We politely clarify that **we did use identical teacher models for our primary comparisons**.
>
> * In Table 1, the baseline denoted `CFT` uses **the exact same teacher (LLaMA3.3-70B Instruct)** as our `CGD` model. Under this strictly controlled setting, CGD clearly outperforms CFT, achieving +5.4% and +7.2% average gains on Math Reasoning, and +4.3% and +3.5% on General Reasoning (e.g., GPQA, MMLU-PRO) for LLaMA3.1-8B and S1.1-3B respectively.
> * **[NEW EVALUATION]** The `CFT with GPT-4o` row was an *additional* reference point trained on official Huggingface CFT data, as GPT-4o is a stronger frontier model. To verify the degradation in Table 2 isn't due to GPT-4o, we evaluated our primary CFT baseline (trained with the **same LLaMA3.3-70B teacher**). It similarly suffered a **severe IFEval degradation** (76.9% to 57.2%), proving format drift is an inherent CFT flaw, not a teacher artifact (aligning with the -21.3% drop for GPT-4o CFT). We will add these results to Table 2.
> * In Table 3, we compared our Qwen2.5-Math-7B CGD model (trained with a **weaker S1.1-32B teacher**, line 305) against the public GPT-4o CFT checkpoint. **Despite this significant disadvantage, CGD achieved a comparable average score (49.0 vs 48.9), a +21.2% absolute gain over base (27.8%)**. Matching a frontier-model teacher provides strong evidence that gains are driven by our training paradigm.
> * **[NEW EXPERIMENT]** To completely remove remaining confounds, we trained a new Qwen2.5-Math-7B CFT baseline using the **exact same S1.1-32B teacher** during this rebuttal. **Result: Under strictly identical conditions, CGD cleanly outperformed the CFT baseline (CGD: 49.0% avg vs. CFT: 46.9% avg).** This definitively isolates the performance gains to our training objective alone. We will add these new results to Table 3 in the camera-ready version as well.
>
> **2. Unconventional Model Family**
> We used both **S1.1** and **Qwen2.5** models. We initially utilized S1.1 due to strict company access policies at the time of research, testing if CGD provides gains even on strong math-specialized models.
> To align with community standards, we later established an external collaboration to validate our method on the standard **Qwen2.5-Math-7B** architecture (Section 4.3, Table 3), where CGD achieved a +22.6% absolute gain.
> Regarding domain overlap inflating S1.1 gains: in Table 1, **both** CFT and CGD were trained on **the exact same S1.1-3B student and S1.1-32B teacher**. Any advantage applies equally to the baseline. Thus, the **+7.2% average math gain of CGD over CFT** is a fair, relative improvement **isolated entirely to the CGD method**.
>
> **3. Evaluation Protocol and Table Captions**
> We apologize for not making the evaluation protocols more clear in the main table captions. As detailed in Appendix A.1, all results in the main tables (Tables 1, 2, 3, 5, 7, 8, 9) evaluate exact match accuracy using greedy decoding (temperature = 0). Regarding Table 4, you are entirely correct that the model name was omitted from the caption. As noted in Appendix D.4.1, this table evaluates the **LLaMA3.1-8B Instruct** model using temp 0.6 / top-p 1.0. We will update all table captions in the final revision to explicitly state the decoding parameters and the specific model being evaluated.
>
> **4. Sensitivity to Critique Quality (Informativeness/Specificity)**
> We agree that studying critique quality beyond just correctness is important. We explicitly analyzed this in **Appendix D.5 (Robustness to Adversarial Critiques)**. In this counterfactual ablation, we deliberately replaced informative critiques with highly specific but *irrelevant/nonsensical* critiques.
>
> * *Result:* The baseline model was derailed by the noise. The CGD model demonstrated "discriminative independence", in which it successfully evaluated the critique's informativeness, ignored the adversarial noise, and solved the problem correctly using its own reasoning. This shows that CGD does not blindly memorize critiques, but learns to verify their informativeness.

---

> > ### Author Rebuttal · Reviewer_zXkr · 2026-04-04
> >
> > As I said, "The ablation on critique correctness mixture (Figure 4) is helpful but focuses on student answer correctness rather than critique informativeness or specificity." There is no other analysis on the critique quality except this single "student answer correctness" metric.

---

> > > ### Author Response · Authors · 2026-04-06
> > >
> > > [`TWO NEW EXPERIMENTS`] We conducted two new analyses: (1) an LLM-as-judge evaluation confirming meaningful quality variation across 20K training critiques, and (2) a controlled training ablation directly testing how critique content affects student performance.
> > >
> > > **Critique quality characterization.** We scored 20K training critiques (10K for incorrect student answers, 10K for correct) using *gpt-oss-120B* on Informativeness (1-5; 5 = identifies the exact error and suggests a fix) and Specificity (1-5; 5 = references specific steps or quantities). Critiques of incorrect answers average **4.36/5** informativeness versus 3.71 for correct-answer critiques, with **42%** of incorrect-answer critiques achieving the maximum score compared to only **4%** for correct-answer critiques. This gap holds across all length quintiles.
> > >
> > > To validate the judge, we scored 600 control critiques (200 per condition): real critiques averaged 4.26, generic phrases ("Try again") scored 1.74, and critiques from unrelated problems scored 1.00, a >2.5-point separation confirming the judge discriminates meaningfully. Together, these results confirm that our training data contains meaningful quality variation along the informativeness and specificity dimensions.
> > >
> > > **Critique content ablation.** Motivated by this quality variation, we constructed three training sets from the same ~10K problems with identical student answers and reference outputs, varying only the critique text. Results averaged over 3 seeds (mean +/- std):
> > >
> > > | Condition | Critique Content | MATH-500 | AMC23 | AIME24 | GSM8K |
> > > |---|---|---|---|---|---|
> > > | A: **Specific & relevant** | Original critiques identifying the exact error | **55.5 +/- 0.6** | **31.7 +/- 1.4** | **12.2 +/- 1.9** | 85.5 +/- 0.2 |
> > > | B: **Specific but irrelevant** | Real critiques from unrelated problems | 53.3 +/- 0.5 | 18.3 +/- 1.7 | 6.7 +/- 0.0 | 85.6 +/- 0.1 |
> > > | C: **Generic** | "The answer is incorrect. Please review..." | 54.0 +/- 0.4 | 26.7 +/- 1.4 | 7.8 +/- 1.9 | 85.7 +/- 0.1 |
> > >
> > > **Key findings:**
> > >
> > > * **Specificity matters (A vs. C):** Specific relevant critiques outperform the generic baseline on reasoning-intensive benchmarks: +1.5 on MATH-500, +5.0 on AMC23, and +4.4 on AIME24. This directly tests the effect of critique specificity, since A and C differ only in how detailed the feedback is. GSM8K shows no separation, as all three conditions already perform near the base model's 85.3%, leaving little room for differentiation on this benchmark.
> > >
> > > * **Relevance matters (B vs. C):** Irrelevant critiques (B) perform *worse* than the generic baseline (C) on AMC23 (18.3 vs. 26.7), indicating the model actively conditions on critique content during training; when that content systematically contradicts the problem, it degrades learning beyond what a content-free placeholder would cause.
> > >
> > > * **Training-time vs. inference-time:** This training-time sensitivity is complementary to the inference-time robustness reported in Appendix D.5: a model already trained on high-quality critiques can discard adversarial noise at test time (Table 16), but training on misleading critiques prevents the model from learning effective reasoning patterns in the first place.
> > >
> > > Combined with the performance drop when critiques are removed entirely (**Figure 3**) and the finding that a CGD model trained on high-quality critiques learns to internally ignore adversarial noise at inference time (**Table 16**), these results establish a clear quality-performance relationship across **five conditions: no critique, noise, generic, irrelevant, and specific**. The attention flow analysis (**Appendix D.3**) further supports this: early layers attend heavily to critiques (48.1% at Layer 0) before pivoting to the problem statement, explaining why content quality, not just critique presence, determines performance. We will include these analyses in the camera-ready version.

---

### Decision · Program_Chairs · 2026-04-30

**Decision:**

Accept (regular)

**Comment:**

CGD addresses a significant bottleneck in model distillation: how to impart deep reasoning and error-correction capabilities without the inference-time latency of multi-turn refinement (e.g., Self-Refine) or the "format drift" (degradation of general instruction-following) caused by training models to generate their own critiques (e.g., Critique Fine-Tuning or CFT).

The core innovation is decoupling critique consumption from generation. During training, the student is conditioned on a flawed initial answer and a teacher's critique to produce a refined final answer. At inference time, the model generates a single-pass response, having internalized the "correction trajectory" during training.

The reviewers unanimously moved toward acceptance following the rebuttal. The consensus is that CGD is a technically sound and practically significant advancement in distillation. By treating critiques as a training-time semantic scaffold, the authors provide a viable path to "thinking" models that remain efficient and versatile.